# Sensing of autoinducer-2 by functionally distinct receptors in prokaryotes

Lei Zhang [1,4✉], Shuyu Li[1,4], Xiaozhen Liu[1], Zhuo Wang[1], Mei Jiang[1], Ruiying Wang[1], Laigong Xie[1], Qinmeng Liu[1], Xiaorong Xie[1], Daohan Shang[1], Mengyun Li[1], Zhiyan Wei[1], Yao Wang[1], Chengpeng Fan[2], Zhao-Qing Luo [3✉] & Xihui Shen [1✉]

Autoinducer-2 (AI-2) is a quorum sensing signal that mediates communication within and between many bacterial species. However, its known receptors (LuxP and LsrB families) are not found in all the bacteria capable of responding to this signaling molecule. Here, we identify a third type of AI-2 receptor, consisting of a dCACHE domain. AI-2 binds to the dCACHE domain of chemoreceptors PctA and TlpQ of *Pseudomonas aeruginosa*, thus inducing chemotaxis and biofilm formation. Boron-free AI-2 is the preferred ligand for PctA and TlpQ. AI-2 also binds to the dCACHE domains of histidine kinase KinD from *Bacillus subtilis* and diguanylate cyclase rpHK1S-Z16 from *Rhodopseudomonas palustris*, enhancing their enzymatic activities. dCACHE domains (especially those belonging to a subfamily that includes the AI-2 receptors identified in the present work) are present in a large number of bacterial and archaeal proteins. Our results support the idea that AI-2 serves as a widely used signaling molecule in the coordination of cell behavior among prokaryotic species.

[1] State Key Laboratory of Crop Stress Biology for Arid Areas, Shaanxi Key Laboratory of Agricultural and Environmental Microbiology, College of Life Sciences, Northwest A&F University, Yangling, Shaanxi 712100, China. [2] Department of Biochemistry and Molecular Biology, School of Basic Medical Sciences, Wuhan University, Wuhan 430071, China. [3] Department of Biological Sciences, Purdue University, West Lafayette, IN, USA. [4] These authors contributed equally: Lei Zhang, Shuyu Li. ✉email: zhanglei0075@nwsuaf.edu.cn; luoz@purdue.edu; xihuishen@nwsuaf.edu.cn

Bacterial quorum sensing (QS) is a cell–cell communication process that is mediated by autoinducers and allows bacteria to coordinate their behaviors in a cell density-dependent manner[1,2]. Whereas the majority of autoinducers such as acyl-homoserine lactones produced by Gram-negative bacteria and oligopeptides secreted by Gram-positive bacteria are dedicated to intraspecies communication[1,3], autoinducer-2 (AI-2) is a well-conserved QS signal that is synthetized by a large cohort of Gram-negative and Gram-positive bacteria and has the capacity to mediate communication at both intra- and interspecies levels[4,5]. Interestingly, AI-2 is not a single signaling molecule but a group of 4,5-dihydroxy-2,3-pentanedione (DPD) derivatives that can convert rapidly to one another (Fig. 1a)[6,7]. DPD is generally synthetized by the enzyme LuxS[1,4], in addition to two non-canonical AI-2 synthesis pathways proposed to be present in some bacteria lacking the *luxS* gene[8]. To date, two AI-2 forms engaged by corresponding bacterial receptors have been identified, including the boron-containing DPD derivative *S*-2-methyl-2,3,3,4-tetrahydroxytetrahydrofuran-borate (*S*-THMF-borate) recognized by LuxP present only in *Vibrio* spp.[6] and the non-borated *R*-2-methyl-2,3,3,4-tetrahydroxytetrahydrofuran (*R*-THMF) recognized by LsrB found in enteric bacteria and some members of several other families (Fig. 1a)[6–13]. Nevertheless, bacterial species possessing the two different types of receptors can communicate with one another via AI-2 signaling due to rapid interconversion between the two active AI-2 forms[2,5].

The AI-2 receptors LuxP and LsrB are both periplasmic binding proteins (PBPs) homologous to ribose binding proteins but share only limited similarity (~11% identity) in their primary sequences[2,9]. LuxP bound to AI-2 converts the activity of the transmembrane sensor histidine kinase (HK) LuxQ from kinase to phosphatase, thus regulating gene expression and changing many density-dependent phenotypes such as bioluminescence, biofilm formation and virulence factor production[1,4,8], whereas the LsrB-AI-2 complex engages the membrane components of the ATP-binding cassette transporter system Lsr to deliver AI-2 into

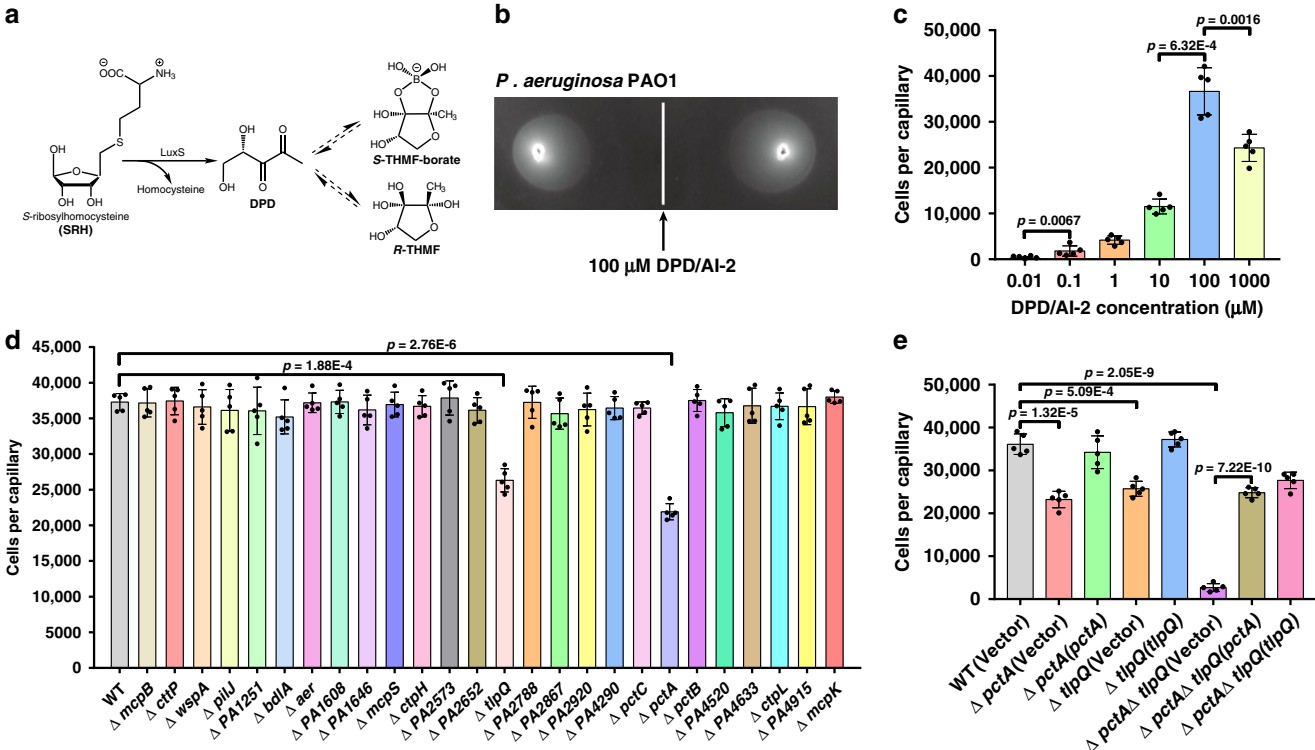

**Fig. 1 Chemotaxis of *P. aeruginosa* PAO1 to AI-2 requires the chemoreceptors PctA and TlpQ. a** A diagram of AI-2 biosynthesis by LuxS[6,7]. The borated AI-2 signal *S*-THMF-borate is recognized by the receptor LuxP in *Vibrio* spp[6], whereas the nonborated *R*-THMF binds to the receptor LsrB found in enteric bacteria and some other microorganisms belonging to the *Rhizobiaceae*, *Bacillaceae* and *Clostridiaceae* families[7-13]. **b** Plate gradient chemotaxis of *P. aeruginosa*. 10 μl aliquots of DPD/AI-2 (100 μM) were spotted onto M9 plates containing 2.5 mM glucose and 0.25% (w/v) Bacto agar and 2 μl aliquots of *P. aeruginosa* cultures with an $OD_{600}$ of 0.6 in M9 medium were placed at 2 cm distance from the DPD/AI-2 spots. The distance from the site of inoculation to the colony edges closest to (D1) and furthest from (D2) the DPD/AI-2 spot was measured and the response index (RI) was calculated as follows: RI = D1/(D1 + D2). Data shown are one representative of five independent experiments with similar results (RI values are presented as mean ± s.d.; $n = 5$ independent experiments). **c** Quantitative capillary chemotaxis induced by AI-2. 230 μl aliquots of *P. aeruginosa* PAO1 with an $OD_{600}$ of 0.1 in chemotaxis buffer were placed into the wells of a 96-well plate and capillaries filled with chemotaxis buffer or different concentrations of DPD/AI-2 solutions (0.01-1000 μM) were immersed into bacterial suspensions for 30 min. Serial dilutions of the contents from the capillaries were plated on LB agar plates and the CFU were determined. Cell numbers were corrected by subtracting the number of cells that have swum into the buffer-containing capillaries. **d** Optimal response to AI-2 requires PctA or TlpQ. Chemotactic responses of the wild-type (WT) PAO1 and 26 mutants to 100 μM DPD/AI-2 were measured by the quantitative capillary chemotaxis assay as described in (**c**). **e** PctA and TlpQ are required for chemotaxis induced by AI-2. The indicated strains were measured for quantitative capillary chemotaxis as described in (**c**). Complementation genes were expressed from derivatives of pME6032. **c–e** Statistical analyses were carried out by results from five independent experiments, each experiment having three technical replicates. Similar results were obtained in five independent experiments and data are presented as mean ± s.e.m. Two-sided, unpaired Student's *t*-test was used for these analyses, and *p* values < 0.05 were considered to indicate statistically significant differences.

cells[9,11]. In addition, LsrB bound to AI-2 is able to drive chemotactic responses in *Escherichia coli* through interactions with the periplasmic sensory domain of the chemoreceptor Tsr[14,15].

Perplexingly, a large number of bacteria robustly responding to AI-2 do not code for LuxP- or LsrB-like AI-2 receptors, raising the possibility that other types of unrecognized receptors for this autoinducer exist[2–4]. For example, in bacterial pathogens such as *Pseudomonas aeruginosa* and *Enterococcus faecalis*, gene expression and phenotypes such as biofilm formation and the production of virulence factors are regulated by AI-2[16–19]. However, earlier efforts to identify potential AI-2 receptors using bioinformatics approaches as well as chemical probes have not been successful[8,13,20], suggesting that alternative strategies may be required to resolve this issue. The inability to identify AI-2 sensors in these AI-2-responsive bacteria have greatly hampered our understanding of the role of AI-2 as a universal signal participating in intra- and interspecies communication[3,4,8,13,20].

Here, by examining a library of *P. aeruginosa* mutants lacking predicted chemoreceptors, we found that PctA and TlpQ sense AI-2 via their periplasmic double CAlcium channels and CHEmotaxis receptors (dCACHE) domains. We also found that AI-2 is recognized by dCACHE domains of the *Bacillus subtilis* HK KinD and the *Rhodopseudomonas palustris* diguanylate cyclase (DGC) rpHK1S-Z16, leading to the induction of their enzymatic activity. More importantly, our bioinformatics analysis suggests that signal transduction proteins harboring a dCACHE domain with the propensity to sense AI-2 are widely distributed in prokaryotes and thus reveals previously unrecognized mechanisms for the detection of and response to AI-2.

## Results

**PctA and TlpQ mediate chemotaxis of *P. aeruginosa* toward AI-2.** Although *P. aeruginosa* does not produce AI-2, it robustly responds to this signaling molecule produced by neighboring heterologous bacteria, leading to changes in the expression of many genes, including those involved in virulence and biofilm formation[16–18]. Yet, how AI-2 stimulates these activities in this bacterium remains elusive. Chemotaxis toward AI-2 has been well demonstrated in *E. coli*[14,15]. Furthermore, both chemosensing and AI-2 signaling have been shown to be implicated in the regulation of *P. aeruginosa* biofilm formation[16–18,21,22]. We thus examined whether *P. aeruginosa* exhibits chemotaxis toward AI-2. By the plate gradient chemotaxis assay, we found that the response index for this bacterium to 100 μM DPD/AI-2 was 0.65 ± 0.02 (Fig. 1b), indicative of chemoattraction[23]. Similarly, DPD/AI-2 induces *P. aeruginosa* chemotaxis at concentrations ranging from 0.1 μM to 1 mM, with an optimal response at 100 μM in the quantitative capillary chemotaxis assay (Fig. 1c).

*P. aeruginosa* is predicted to encode a complex chemosensory network that consists of at least 26 chemoreceptors working with four chemosensory pathways[21]. To identify the chemoreceptors potentially involved in AI-2 chemotaxis, we created a series of mutants by deleting each of these 26 chemoreceptor genes and examined their chemotactic response to AI-2. Whereas most of these mutations did not affect the response of *P. aeruginosa* to AI-2, deletion of *pctA* or *tlpQ* significantly reduced chemotaxis to this compound (Fig. 1d). Expression of *pctA* and *tlpQ* in the corresponding mutants fully restored their chemotaxis toward AI-2. Furthermore, mutants lacking both *pctA* and *tlpQ* have completely lost the ability to respond to AI-2 (Fig. 1e). These observations indicate that PctA and TlpQ are essential for the chemotaxis of *P. aeruginosa* toward AI-2.

**Identification of PctA and TlpQ as AI-2 receptors.** Sequence analysis indicates that both PctA and TlpQ harbor a ligand-binding

domain (LBD) of the dCACHE family, a structure with two extra-cytoplasmic PAS-like subdomains termed membrane-proximal and membrane-distal modules, respectively[24,25]. Although PctA and TlpQ share only ~17% sequence identity in the LBD regions (Supplementary Fig. 1), alignment of the 3D structures of PctA-LBD (PDB ID: 5LTX)[26] and TlpQ-LBD (PDB ID: 6FU4)[24] using TM-align[27] suggests that they are mostly in the same fold (TM-score = 0.79, Supplementary Fig. 2). Both PctA and TlpQ are known to drive chemotactic responses by directly binding to specific ligands[24,28,29]. Whereas PctA-LBD appears to exclusively bind amino acids[28,29], TlpQ-LBD specifically recognizes histamine and polyamines[24]. Despite the well-established ligands for PctA-LBD and TlpQ-LBD, the results from chemotaxis tests (Fig. 1b–e) led us to speculate that these two LBDs may interact with AI-2. We first determined the ability of signal released from recombinant proteins purified from an *E. coli* strain capable of AI-2 production[10] to stimulate light production in the *Vibrio harveyi* strain MM32 lacking *luxN* and *luxS*[7]. Upon denaturing by heat treatment, purified LBDs of PctA and TlpQ from the *luxS*+ *E. coli* strain released ligands capable of inducing light production in strain MM32 at levels comparable to those by LsrB, an established AI-2-binding protein[10], whereas these two LBDs purified from an *E. coli* strain lacking *luxS* did not release ligands that detectably induced bioluminescence in strain MM32 (Fig. 2a). In contrast, the dCACHE-type LBDs of PctB and PctC, two paralogs of PctA in *P. aeruginosa*, purified from the *luxS*+ *E. coli* strain released no detectable AI-2 activity upon heat denaturation (Fig. 2a). These results suggest that the LBDs of PctA and TlpQ, but not PctB or PctC, have the capacity to bind AI-2.

We further determined the binding affinity between AI-2 and these two receptors by isothermal titration calorimetry (ITC) and found that it has a disassociation constant ($K_d$) of 0.14 ± 0.02 μM and 0.12 ± 0.01 μM for PctA-LBD and TlpQ-LBD, respectively (Fig. 2b, c), which are comparable to the $K_d$ values of AI-2 for such established receptors as LuxP (0.16 μM)[30] and LsrB (0.19–0.81 μM)[10]. Under the same experimental conditions, low-affinity binding of AI-2 to PctB-LBD (181 ± 17 μM) and PctC-LBD (99 ± 13 μM) was detected (Supplementary Fig. 3a, b). These results establish that AI-2 specifically binds to PctA-LBD and TlpQ-LBD with high affinity. In culture supernatants of bacteria such as *Yersinia pestis*, AI-2 concentrations could reach micromolar levels[31]. Under natural niches with relatively scarce nutrient that harbor multiple bacterial species, the concentrations of AI-2 likely are considerably lower than those seen in pure cultures. Nevertheless, the $K_d$ values presented here suggest that this signaling molecule is a physiologically relevant ligand for PctA and TlpQ.

Using *S*-adenosylmethionine (SAM) as the methyl donor, the methyltransferase CheR converts specific glutamate residues in the cytoplasmic signaling domain of the methyl-accepting chemotaxis protein (MCP) to glutamyl methyl esters[22,32]. Amino acid sequence alignment of PctA and the chemoreceptor Tsr from *E. coli* predicted E381, E395 and E614 are potential methylation sites in PctA (Supplementary Fig. 4). Given that CheR1-catalyzed methylation of PctA is modulated by its amino acid ligands[32], we investigated whether AI-2 affects PctA methylation by CheR1. Under our experimental conditions, CheR1 did not detectably alter the methylation at E395 and E614 (Supplementary Fig. 5). In contrast, PctA can be methylated at E381 by CheR1 and such modification was induced by DPD/AI-2 (Fig. 2d). These results further suggest that AI-2 is a ligand for PctA.

**AI-2 induces biofilm formation in *P. aeruginosa* via PctA and TlpQ.** AI-2 is known to regulate biofilm formation by *P. aeruginosa* in a dose-dependent manner[18], but the mechanism of such

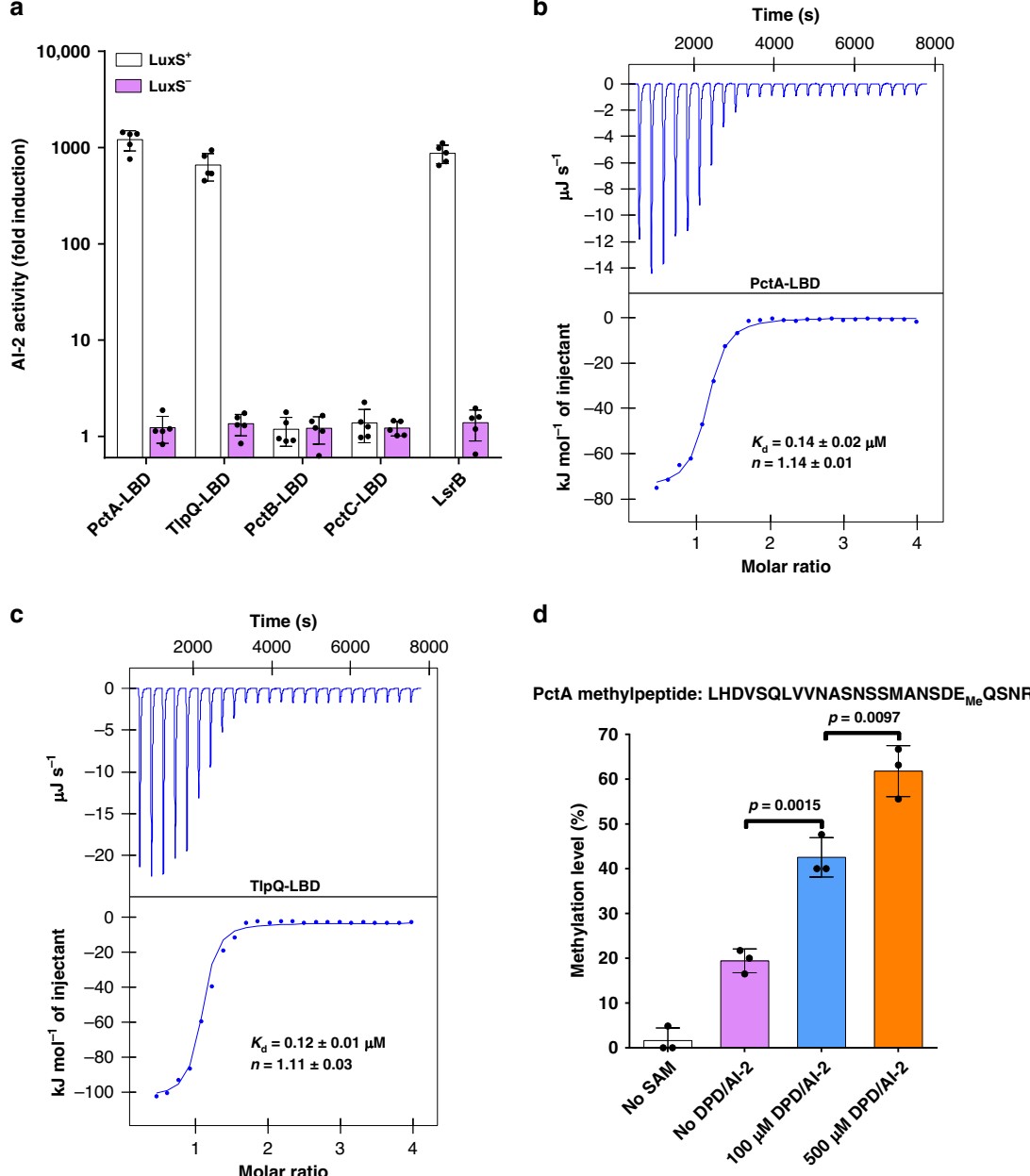

**Fig. 2 AI-2 is a ligand for the chemoreceptors PctA and TlpQ of *P. aeruginosa*. a** PctA-LBD and TlpQ-LBD are capable of retaining AI-2. Proteins were expressed in a *luxS*⁺ (white bars) or *luxS*⁻ (purple bars) *E. coli* strain and light production by the AI-2 reporter *V. harveyi* strain MM32 (*luxN*⁻, *luxS*⁻) was measured following the addition of a buffer control or ligands released from the purified proteins upon denaturing by heating. LsrB from *E. coli* was used as a positive control, and the LBDs of PctB and PctC was used as negative controls. AI-2 activity is reported as fold induction of bioluminescence over background obtained in the buffer control alone (mean ± s.e.m.; $n = 5$ independent experiments). **b, c** PctA-LBD and TlpQ-LBD interact with AI-2. The binding affinity was evaluated using ITC analysis. ITC data and plots of injected heat for injections of DPD/AI-2 solution (700 μM) into the sample cell containing 70 μM PctA-LBD (**b**) or TlpQ-LBD (**c**) are shown in the upper and lower plots, respectively. A control experiment, in which DPD/AI-2 solution (700 μM) was injected into the buffer in the sample cell, was performed (Supplementary Fig. 15a) and heats of dilution were used to correct the data. The binding curves corrected for the dilution effects were fit to a one-site binding model. Data shown are one representative of three independent experiments with similar results. The $K_d$ and binding stoichiometry ($n$) were calculated by the NanoAnalyze software and presented as mean ± s.d. of three independent experiments. **d** PctA methylation at E381 by CheR1 is induced by AI-2. Methylation of PctA by CheR1 was carried out by co-incubating with SAM in the presence or absence of DPD/AI-2. Reactions not receiving SAM were established as controls. Methylation of PctA at E381 was quantified by liquid chromatography-tandem mass spectrometry (LC-MS/MS) analysis. Data are mean ± s.e.m. of three independent experiments. *P* values were determined by two-tailed unpaired Student's *t*-test, and differences were considered statistically significant at $p < 0.05$.

regulation is unknown. Because methylation-dependent chemotaxis plays an important role in cell attachment and biofilm formation by *P. aeruginosa*[22], we hypothesized that AI-2 regulates this process via PctA and TlpQ. Indeed, the inclusion of 100 nM

DPD/AI-2 in cultures of strain PAO1 resulted in a significant increase in biofilm formation (Fig. 3a). Such induction was impaired in mutants Δ*pctA* and Δ*tlpQ*, and was completely abolished in the double mutant Δ*pctA*Δ*tlpQ* (Fig. 3a). Moreover,

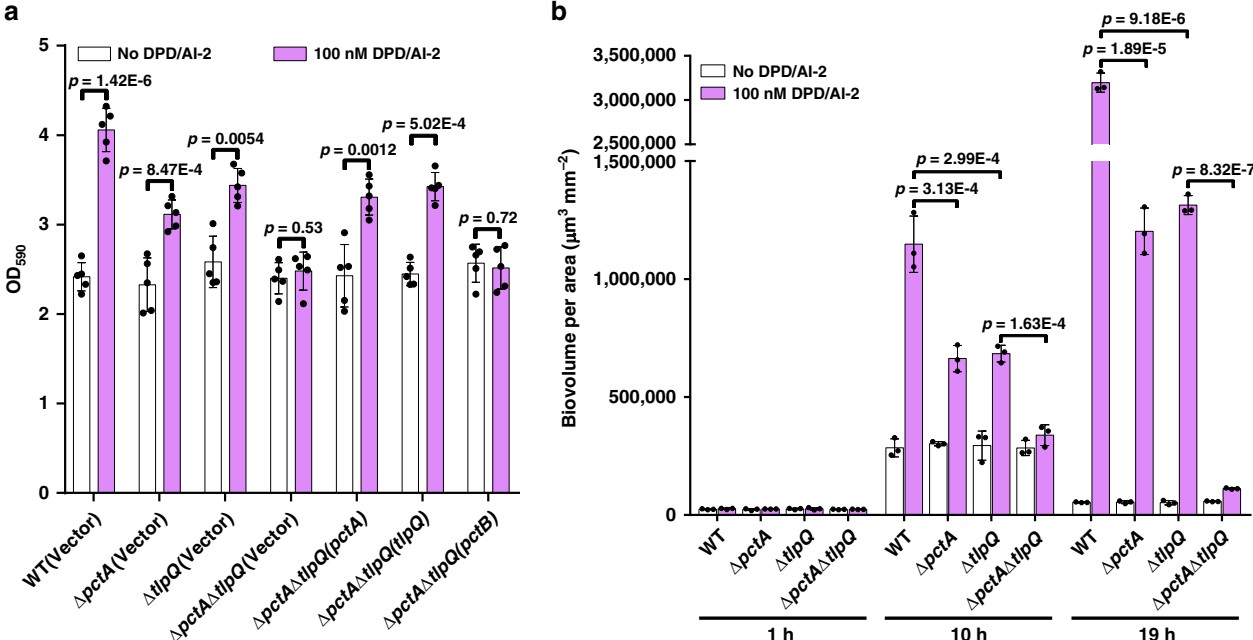

**Fig. 3 AI-2 regulates *P. aeruginosa* biofilm formation via PctA and TlpQ. a** Crystal violet quantification of biofilm formation by *P. aeruginosa* strains in the presence or absence of DPD/AI-2. 190 μl aliquots of *P. aeruginosa* strains with an $OD_{600}$ of 0.05 in TSB medium were placed into the wells of a 96-well plate and then 10 μl aliquots of DPD/AI-2 (2 μM) or a buffer control were added to the wells. Biofilms were stained with crystal violet and quantified using optical density measurement after incubation at 37 °C for 48 h. Data are mean ± s.e.m. of five independent experiments. **b** Quantification of biofilm formation by *P. aeruginosa* strains from confocal imaging. Confocal dishes were inoculated with 190 μl of mCherry-labeled *P. aeruginosa* strains diluted to an $OD_{600}$ of 0.01 in TSB medium and 10 μl of the DPD/AI-2 solution (2 μM) or a buffer control, and biofilms formed were detected by confocal laser-scanning microscopy after incubation at 37 °C for 1, 10 and 19 h, respectively. Images were reconstructed using the Imaris 9.0 software package (Bitplane, AG) (Supplementary Fig. 6) and biofilm biovolumes were quantified using COMSTAT (www.comstat.dk). Biovolumes were calculated from three biological replicates and each biological replicate was derived from an average of five confocal images. Similar results were obtained in three biological replicates and data are presented as mean ± s.e.m. **a**, **b** Statistical significance was evaluated using two-tailed unpaired Student's *t*-test. *P* values < 0.05 were considered to indicate statistically significant differences. WT, wild-type.

expression of *pctA* or *tlpQ*, but not *pctB* in Δ*pctA*Δ*tlpQ* restored DPD/AI-2-induced biofilm formation at levels similar to that of Δ*pctA* or Δ*tlpQ* (Fig. 3a). When the wild-type strain PAO1 and mutants Δ*pctA*, Δ*tlpQ* and Δ*pctA*Δ*tlpQ* were labeled with mCherry and the biofilm biovolume for each was measured by confocal laser-scanning microscopy, PctA- and TlpQ-dependent AI-2-induced biofilm formation was similarly observed (Fig. 3b and Supplementary Fig. 6). Thus, AI-2 induces biofilm formation in *P. aeruginosa* by directly engaging PctA and TlpQ.

**PctA-LBD and TlpQ-LBD interact with the nonborated form of AI-2.** The two forms of AI-2 (*S*-THMF-borate and *R*-THMF) can interconvert and addition of borate is known to shift the equilibrium of AI-2 molecules toward the *S*-THMF-borate form[7,10]. We thus investigated the role of boron in the binding of AI-2 to PctA-LBD and TlpQ-LBD. When the products from the in vitro reaction of *S*-adenosylhomocysteine (SAH) with methylthioadenosine/SAH nucleosidase (Pfs) and LuxS in a borate-depleted system were titrated into PctA-LBD in borate-depleted buffer, we detected a binding affinity ($K_d$) of 26 ± 4 nM (Fig. 4a). Moreover, the addition of 150 μM boric acid in the reaction products and the protein solution led to a 119-fold decrease in binding affinity ($K_d$ = 3.1 ± 0.1 μM) (Fig. 4b). Similarly, boric acid also weakened the interactions between the products of the Pfs/LuxS reaction and TlpQ-LBD (Supplementary Fig. 7a, b). These results suggest that the nonborated form of AI-2 is the preferred ligand for PctA-LBD and TlpQ-LBD.

A pocket in the membrane-distal module of both PctA-LBD and TlpQ-LBD is involved in ligand recognition[24,26,28]. Calculation of the binding isotherm data showed that AI-2 binds to PctA-LBD and TlpQ-LBD in a 1:1 stoichiometry (*n* = 0.93–1.29 sites) (Fig. 2b, c, Fig. 4a, b and Supplementary Fig. 7a, b), suggesting that the pocket in the membrane-distal module of these two proteins is also the binding site for AI-2. By Glide extra precision (XP) docking analysis[33], we obtained several binding modes of *R*-THMF in the amino acid-binding pocket of PctA-LBD (PDB ID: 5T7M)[26]. Among these, the best conformation has the lowest docking score of −7.82 kcal mol⁻¹, which is higher than those (ranging from −13.74 to −11.53 kcal mol⁻¹) obtained from the docking of L-Met, L-Ile or L-Trp to PctA-LBD[26] but is below the lower threshold value of −6 kcal mol⁻¹ for significance[26]. This conformation suggests that AI-2 makes close contact with Y101, M111, Y121, R126, W128, Y144, D146, A147, and D173 in the active pocket (Fig. 4c). In support of this binding model, mutations in each of these residues resulted in marked reduction in AI-2 binding affinity for PctA-LBD (Fig. 4d). In comparison, the best conformation of *S*-THMF-borate bound in the active pocket of PctA-LBD has the lowest docking score of −2.92 kcal mol⁻¹, which is distant from the threshold for significance and thus suggests a weak interaction between these two molecules. When both forms of AI-2 were docked into the histamine-binding pocket of TlpQ-LBD (PDB ID: 6FU4)[24], all binding conformations of *S*-THMF-borate give positive docking scores, suggesting unfavorable binding of *S*-THMF-borate to TlpQ-LBD. In contrast, the best conformation of *R*-THMF bound in the active pocket of TlpQ-LBD (Supplementary Fig. 8a) has the

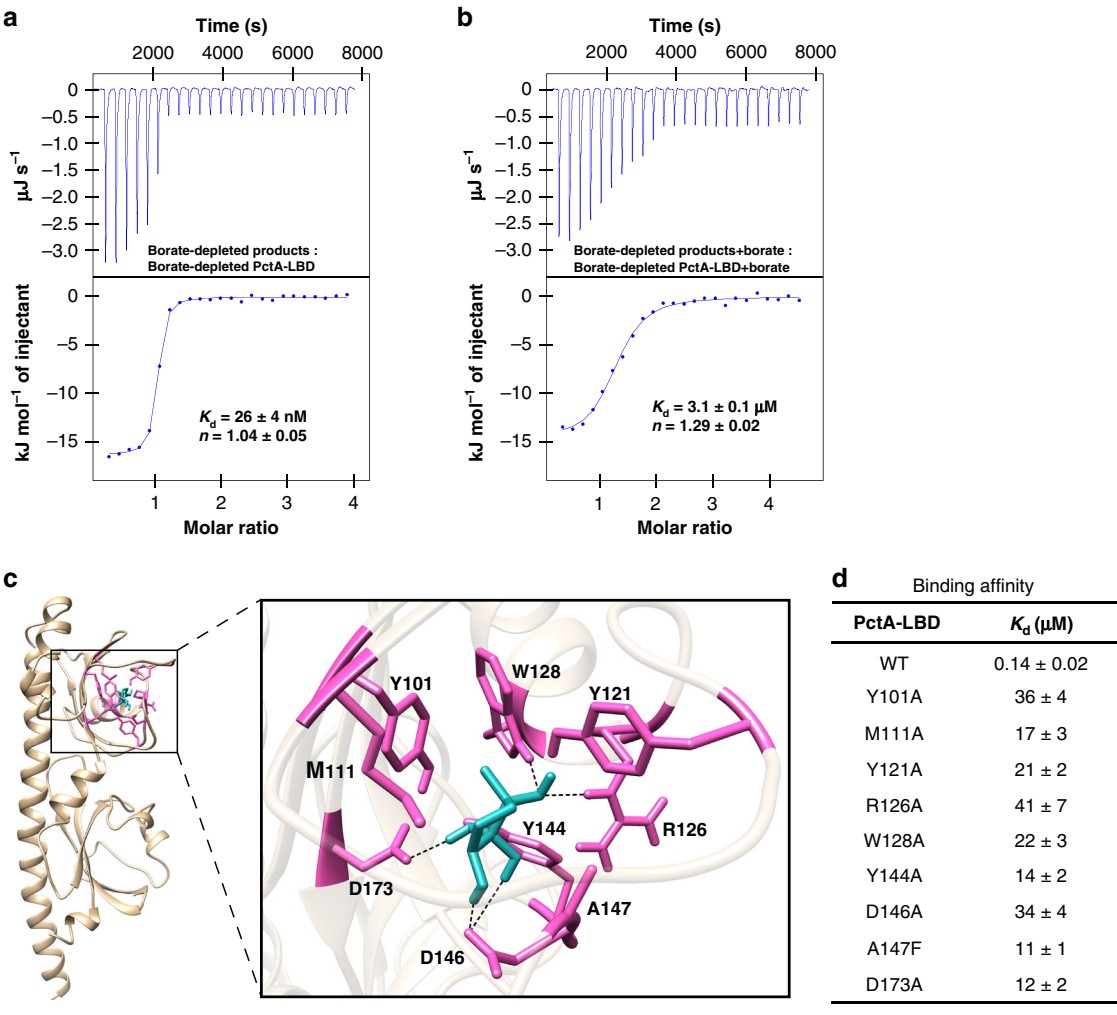

**Fig. 4 The dCACHE domain of PctA prefers the nonborated form of AI-2. a, b** Higher binding affinity of the products of the Pfs/LuxS reaction to PctA-LBD is detected under borate-depleted conditions. The binding affinity was evaluated using ITC analysis. In vitro reaction of SAH with Pfs and LuxS proteins was performed with plasticware and borate-depleted water, and concentration of DPD/AI-2 in the reaction products is ~13 μM. Proteins were dialyzed against borate-depleted buffer before use. ITC data and plots of injected heat for automatic injections of the borate-depleted reaction products (**a**) or the borate-depleted products supplemented with 150 μM boric acid (**b**) into the sample cell containing 1.3 μM borate-depleted PctA-LBD (**a**) or borate-depleted PctA-LBD supplemented with 150 μM boric acid (**b**) are shown in the upper and lower plots, respectively. Microcalorimetric data were corrected by subtracting the heats of dilution for ligand solutions injected into buffer (Supplementary Fig. 15b, c). Three independent ITC experiments were performed and similar results were obtained. The $K_d$ and binding stoichiometry ($n$) calculated by the NanoAnalyze software from three independent experiments are presented as mean ± s.d. **c** Predicted binding mode of $R$-THMF in the amino acid-binding pocket of PctA-LBD. PctA-LBD (PDB ID: 5T7M)[26] and $R$-THMF extracted from $R$-THMF-LsrB (PDB ID: 1TJY)[7] were used for Glide XP docking analysis. The conformation with the lowest docking energy is given via the Chimera software. The key residues of PctA-LBD involved in $R$-THMF binding are shown as purple sticks, and $R$-THMF is shown as cyan sticks. Five potential hydrogen bonds are indicated by dashed lines. **d** Binding of DPD/AI-2 (700 μM) to PctA-LBD and its mutants (70 μM) under normal conditions. The binding affinity was evaluated using ITC analysis. Data shown are mean ± s.d. of three biological replicates. WT, wild-type.

lowest docking score of $-5.70$ kcal mol$^{-1}$, which is comparable to that ($-6.58$ kcal mol$^{-1}$) of histamine-TlpQ-LBD docking. Furthermore, mutations in W192, Y208, D210 and D239, which lie closest to the ligand (<2.7 Å), drastically reduced the binding affinity of TlpQ-LBD for AI-2 (Supplementary Fig. 8b). Collectively, these results suggest that PctA-LBD and TlpQ-LBD interact with the nonborated form of AI-2, likely $R$-THMF, through the conserved ligand-binding pockets within their membrane-distal modules.

**Identification of PctA-LBD homologues that bind AI-2.** The dCACHE domain is predicted to serve as an extracytoplasmic sensory module for all major types of signal transduction proteins

in prokaryotes[24,25]. We explored the role of this domain in AI-2 sensing by using PctA-LBD as a query to search the PDB_mmCIF70 database by HHpred[34], which allowed the retrieval of 19 bacterial dCACHE domains highly similar in their structures (>98.5% probability) (Supplementary Fig. 9). Among these, the LBD domains from two additional chemoreceptors, two sensor HKs, and two putative DGCs were examined for AI-2 binding by the ability of the signal molecule released from recombinant proteins purified from the $luxS^+$ E. coli strain to induce bioluminescence in V. harveyi strain MM32[7]. In the V. harveyi MM32 reporter assay, AI-2 binding activity was observed for the LBD of HK KinD from B. subtilis and of the putative DGC rpHK1S-Z16 from R. palustris (Fig. 5a). In contrast, no such activity was detected in the predicted LBD domain from

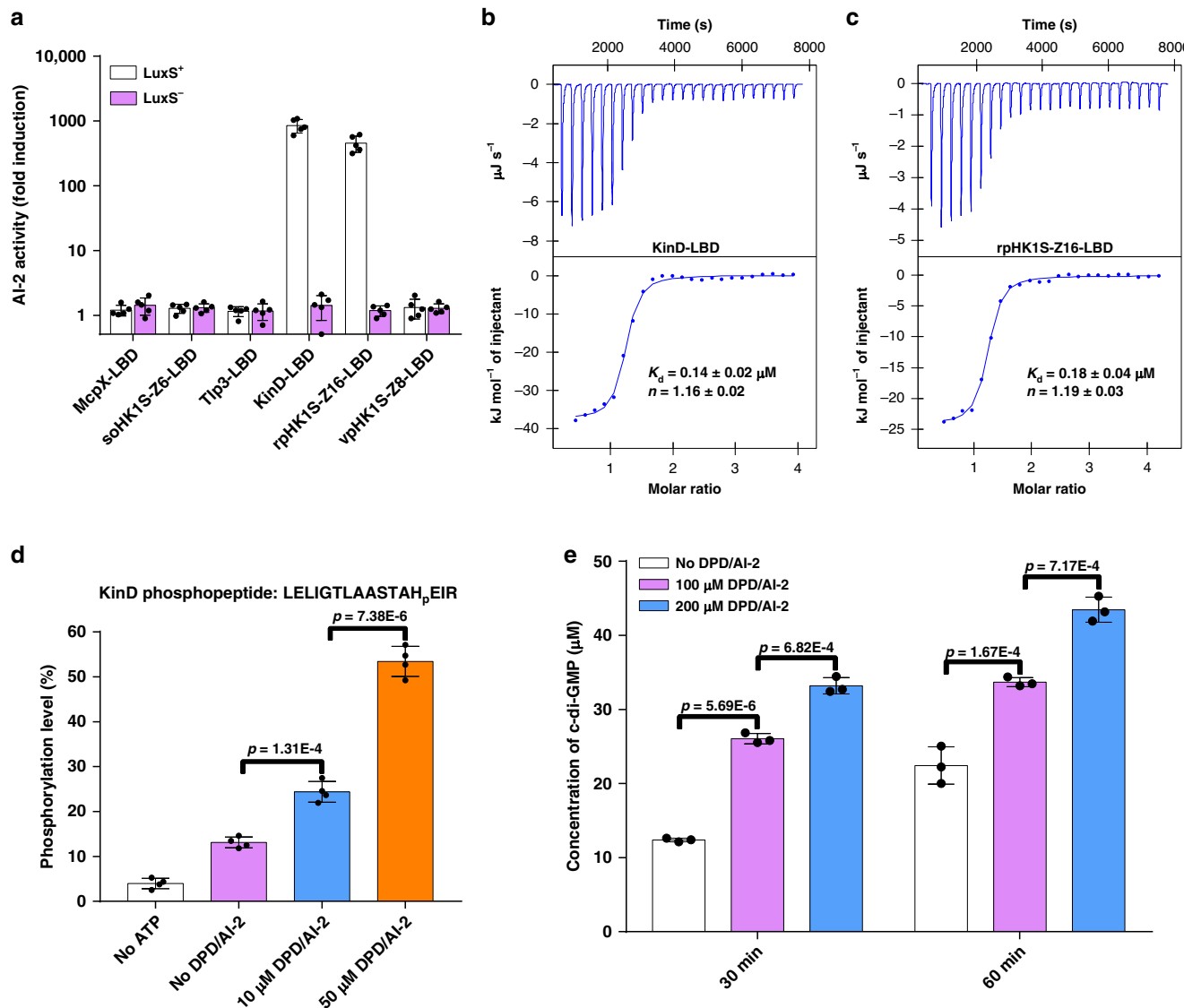

**Fig. 5 AI-2 induces the activity of KinD from *B. subtilis* and of DGC rpHK1S-Z16 from *R. palustris*. a** The LBD of KinD from *B. subtilis* and rpHK1S-Z16 from *R. palustris* is capable of retaining AI-2. Bioluminescence in *V. harveyi* MM32 was induced by addition of ligands released from purified proteins expressed in a *luxS⁺* (white bars) or *luxS⁻* (purple bars) *E. coli* strain. Results are shown as fold induction relative to the light production induced by a buffer control (mean ± s.e.m.; $n = 5$ independent experiments). **b, c** AI-2 specifically binds to the LBDs of KinD and rpHK1S-Z16 with high affinity. The binding affinity was evaluated using ITC analysis. The upper panels show the sequential heat pulses for 700 μM DPD/AI-2 injected into 70 μM KinD-LBD (**b**) or rpHK1S-Z16-LBD (**c**), and the lower panels show the integrated data that were corrected for heat of dilution of the ligands into buffer and fit to a one-site binding model. Data shown are one representative of three independent experiments with similar results. The $K_d$ and binding stoichiometry (*n*) were calculated using the NanoAnalyze software and presented as mean ± s.d. of three independent experiments. **d** AI-2 stimulates the autokinase activity of KinD. Autophosphorylation of KinD was carried out in reactions with or without DPD/AI-2. Reactions without ATP were established as controls. KinD phosphorylation was detected by determining the ratio between phosphopeptide and total peptides using LC-MS/MS. Data are mean ± s.e.m. of four independent experiments. **e** AI-2 induces the DGC activity of rpHK1S-Z16. Membrane fractions containing rpHK1S-Z16 were incubated with GTP in the presence or absence of DPD/AI-2 at 30 °C for 0, 30, and 60 min and the product was analyzed by HPLC (Supplementary Fig. 10). The level of c-di-GMP was determined from a standard curve established with a serially diluted c-di-GMP solution. Data are mean ± s.e.m. of three independent experiments. **d, e** *P* values were determined using the two-tailed unpaired Student's *t*-test. $p < 0.05$ was considered to indicate a statistically significant difference.

chemoreceptors McpX from *Sinorhizobium meliloti*, Tlp3 from *Campylobacter jejuni*, the sensor HK soHK1S-Z6 from *Shewanella oneidensis* or the putative DGC vpHK1S-Z8 from *Vibrio parahaemolyticus* (Fig. 5a). Binding analysis by ITC showed that the LBD of KinD and rpHK1S-Z16 binds AI-2 with $K_d$ values of 0.14 ± 0.02 μM and 0.18 ± 0.04 μM, respectively (Fig. 5b, c). More importantly, DPD/AI-2 was able to induce the autokinase activity of KinD (Fig. 5d) and the DGC activity of rpHK1S-Z16 in c-di-GMP synthesis (Fig. 5e and Supplementary Fig. 10). These results

indicate that AI-2 is also a ligand for the HK KinD from *B. subtilis* and the DGC rpHK1S-Z16 from *R. palustris*.

**Proteins with the dCache_1 domain capable of sensing AI-2 are present in diverse bacteria and archaea.** The sensory domains of PctA, TlpQ, KinD, and rpHK1S-Z16 are all categorized into dCache_1, the largest subfamily of the dCACHE family[25]. We expanded our search of the distribution of the

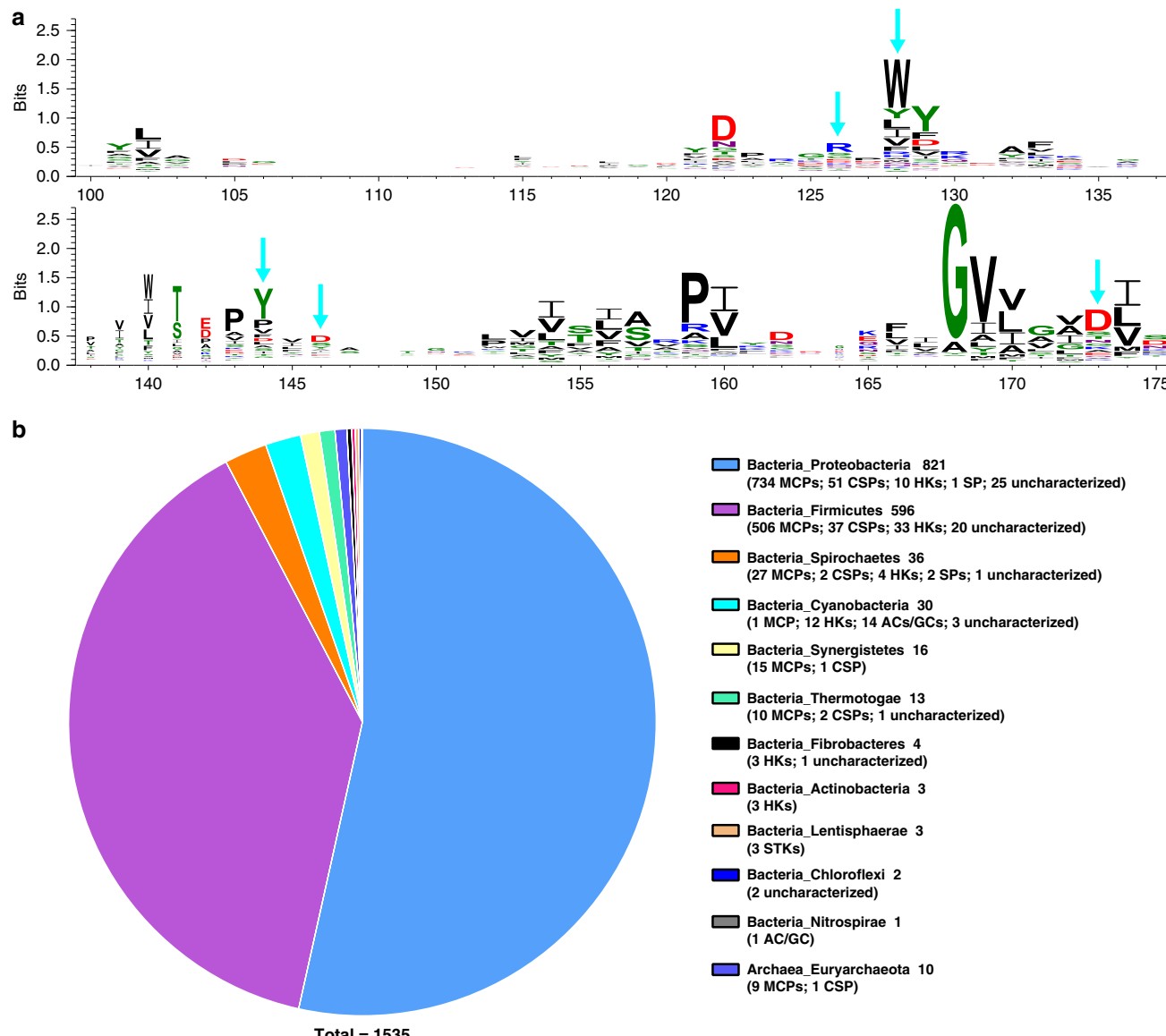

**Fig. 6 Widespread occurrence of dCache_1-containing AI-2 receptors in bacteria and archaea. a** A conservation pattern found in dCache_1 domains. Multiple alignment analysis of 18970 dCache_1 domains was performed with ClustalW embedded in MEGA7 (Supplementary Data 3). The region corresponding to the active pocket of PctA-LBD (residues 100–175) is illustrated using WebLogo 3. Cyan arrows above the WebLogo denote the five highly conserved positions corresponding to R126, W128, Y144, D146, and D173 of PctA. **b** Distribution of 1535 dCache_1-containing proteins with all the five conserved residues corresponding to R126, W128, Y144, D146, and D173 of PctA at the phylum level in prokaryotes. The number of occurrences for each type of signal transduction proteins is listed in parentheses.

dCache_1 domain potentially involved in sensing AI-2 by using the Pfam (version 32.0) and identified 18970 proteins (Supplementary Data 1). Alignment of the dCache_1 domain from these proteins revealed that the five conserved residues R126, W128, Y144, D146 and D173 found in the active pocket of PctA-LBD[26,28] are present at the highest frequencies (Fig. 6a). 1535 dCache_1 domains with all the five conserved residues were found (Supplementary Data 2). Among these, 1482 can potentially function as extracytoplasmic sensors of MCPs, HKs, c-di-GMP synthases and phosphodiesterases (CSPs), serine phosphatases (SPs), serine/threonine kinases (STKs), adenylate- or guanylate cyclases (ACs/GCs), 1 is from an uncharacterized protein with cyclic nucleotide-binding domain and 52 are from completely uncharacterized proteins without any predicted functional domains or motifs (Fig. 6b and Supplementary

Data 2). To examine the ability of these sensory domains to bind AI-2, we randomly selected 17 from bacteria and 2 from archaea predicted to function as sensors of MCPs, HKs, CSPs, SPs, STKs or ACs/GCs and prepared recombinant proteins from the *luxS*⁺ *E. coli* strain. All of these 19 proteins detectably retained AI-2, suggestive of their ability to sense this signaling molecule (Supplementary Fig. 11). Of note, although not all five conserved residues are present in the dCache_1 domains of TlpQ, KinD and rpHK1S-Z16 (Supplementary Fig. 12), each of these proteins was found to bind AI-2, suggesting the existence of yet unrecognized AI-2-binding motifs. Taken together, our results suggest that the extracytoplasmic dCache_1 domains of various signal transduction proteins constitute a large group of widely distributed AI-2 receptors in both bacteria and archaea.

## Discussion

The *luxS* gene responsible for the biosynthesis of AI-2 is widespread throughout the bacterial kingdom[1,4]. Many bacteria lacking a *luxS* ortholog respond to AI-2 but the mechanism for the detection and response remains elusive, largely owing to the lack of information about its receptors[3,13,20]. Here we resolved this conundrum by identifying three types of dCache_1 domain-containing signal transduction proteins, including two MCPs, one HK and one DGC, as AI-2 receptors from both Gram-negative (*P. aeruginosa* and *R. palustris*) and Gram-positive (*B. subtilis*) bacteria. Furthermore, our results suggest that more than 1500 dCache_1-containing proteins predicted to function as MCPs, HKs, CSPs, SPs, STKs or ACs/GCs from bacteria and archaea are potential AI-2-binding receptors. These findings provide new mechanistic insights into how AI-2 functions as a signal for intraspecies and interspecies communication. The identification of dCache_1-containing signal transduction proteins from diverse bacteria and archaea as AI-2 receptors has greatly expanded our understanding of AI-2 as a potentially "universal" signal in communication among prokaryotic species that co-occupy niches of environmental or medical importance.

Bacterial transmembrane receptors with periplasmic sensory domains can detect extracellular signals by either binding signals directly or interacting with PBPs in complex with signaling molecules[1,24]. Whereas the two known AI-2 receptors LuxP and LsrB are both PBPs that transmit AI-2 signals into cells via their respective transmembrane receptors[6,7], the novel AI-2 receptors identified here are themselves transmembrane proteins with a dCache_1 domain, which is amongst the most abundant extra-cytoplasmic sensors in bacteria[25,35] and is also present in a small number of archaea and eukaryotes (Supplementary Data 1). dCache_1 domains have been found in all major types of signal transduction proteins in prokaryotes, including members of the MCP, HK, CSP (GGDEF, EAL, HD-GYP), SP (SpoIIE), STK, and AC/GC families[25]. Whereas most known ligands that directly bind the dCache_1 domain are amino acids[26,28,29], some LBDs of this family are found to recognize such diverse compounds as histamine, polyamines[24], quaternary amines[36], organic acids[37], purines[38], galactose[35], and thiamine[39]. Moreover, some dCache_1 domains appear to recognize a range of different types of ligands. For example, the dCache_1-containing chemoreceptor Tlp3 of *Campylobacter jejuni* senses multiple ligands including amino acids, organic acids, purine and thiamine[39]. To our knowledge, however, this is the first report identifying various signaling proteins harboring a dCache_1 domain as AI-2 receptors, thus further expanding the range of ligands recognized by the dCache_1 family.

Previous studies using high-throughput ligand screening experiments with commercially available ligand collections have suggested that PctA-LBD specifically binds amino acids[29] whereas TlpQ-LBD specifically binds histamine and polyamines[24]. Our results have expanded the ligand-binding capacity of these two dCache_1-type LBDs to include AI-2 (Fig. 2). Intriguingly, the AI-2-binding dCache_1 domains with the five conserved residues corresponding to R126, W128, Y144, D146, and D173 of PctA are present in proteins of both bacteria and archaea (Supplementary Data 2 and Supplementary Fig. 11), suggesting that this type of dCache_1 domains evolved from the same ancestor. Interestingly, the AI-2-binding PctA has been proposed to be the ancestor of PctB and PctC[26], both of which contain four of the five conserved residues in their dCache_1 domains (Supplementary Fig. 12). The variation in one of the five conserved residues may cause PctB-LBD and PctC-LBD to have lower affinities for AI-2. In support of this speculation, mutations of the non-conserved residue to conserved residue within PctB-LBD (E146D) and PctC-LBD (F147Y) increased their AI-2 binding affinity (0.21-0.24 μM) (Supplementary Fig. 3c, d), which became comparable to that of PctA-LBD (0.14 μM) (Fig. 2b). Nevertheless, the AI-2-binding TlpQ has been predicted to be evolutionarily distinct from the three paralogous chemoreceptors PctA, PctB and PctC[26], while the AI-2-binding LBDs of KinD and rpHK1S-Z16 possess none of the five conserved residues (Supplementary Fig. 12), suggesting the existence of AI-2-binding motifs of other origins.

The identification of dCache_1-containing AI-2 receptors in both AI-2-producing bacteria such as *B. subtilis*[13] and non-AI-2-producing bacteria such as *P. aeruginosa*[4] establishes that this type of AI-2 receptors mediate both intraspecies and interspecies communication among AI-2-producing and non-AI-2-producing bacteria. Furthermore, the presence of the dCache_1-containing AI-2 receptors in archaea (Supplementary Fig. 11) suggests that AI-2 can be used for inter-kingdom crosstalk between bacteria and archaea.

AI-2-mediated interspecies communication has been shown to coordinate such important phenotypes as coaggregation, biofilms and virulence[16–18,40,41]. For example, AI-2 produced by *Enterococcus faecalis* leads to enhanced aggregation and biofilm formation by *E. coli* as well as coaggregation of these two species[14,41]. Patients of cystic fibrosis and other disease conditions are often co-infected by *P. aeruginosa* and AI-2-producing bacterial species such as *Staphylococcus aureus, Klebsiella pneumoniae* and *Streptococcus mitis*[17,18,40]. The observation that PctA and TlpQ facilitate the response of *P. aeruginosa* toward AI-2 may explain why AI-2 in cystic fibrosis lungs enhances the virulence of *P. aeruginosa*[16–18,40]. Thus, compounds capable of interfering with signaling mediated by AI-2 represent a novel intervention strategy for infections caused by multiple bacterial pathogens, and our findings have laid the foundation for future screening, design and optimization of such agents.

## Methods

**Bacterial strains, plasmid constructions, and growth conditions**. Strains and plasmids are listed in Supplementary Table 1 and primers are listed in Supplementary Table 2. *P. aeruginosa* strain PAO1 and its derivatives used in this study were usually grown at 37 °C in either Luria-Bertani (LB) or tryptic soy broth (TSB) medium, unless specified otherwise. *V. harveyi* MM32 was grown in AB medium[42] at 30 °C. *R. palustris* was grown in TSB, and *S. meliloti, S. oneidensis, V. parahaemolyticus, B. subtilis* and *E. coli* strains were grown in LB. The DNA fragments encoding dCache_1 domains of 17 signal transduction proteins were synthesized by Genewiz (Suzhou, China). To express and purify soluble GST- and His₆-tagged recombinant proteins, genes were cloned into pGEX-6P-1 and pET-28a, and then transformed into *E. coli* XL1-Blue and BL21(DE3) host strains, respectively. When required, antibiotics were used at the following concentrations: ampicillin, 100 μg ml⁻¹; kanamycin, 50 μg ml⁻¹; chloramphenicol, 20 μg ml⁻¹; gentamicin, 15 μg ml⁻¹ for *E. coli* and 150 μg ml⁻¹ for *P. aeruginosa*; tetracycline, 15 μg ml⁻¹ for *E. coli* and 160 and 200 μg ml⁻¹ for *P. aeruginosa* during growth in liquid cultures and on plates, respectively.

In-frame deletion mutants of *P. aeruginosa* were constructed by double-crossover allelic exchange using derivatives of the suicide vector pK18mobsacB harboring gentamicin resistance cassette (GMC) and the 5′ region and 3′ region of target genes[43]. The GMC was amplified from plasmid p34S-Gm and inserted into the pK18mobsacB vector. The 5′ and 3′ flanking regions of the gene of interest were amplified separately and ligated together by overlap extension PCR. After digesting the DNA products overlapped by PCR and the pK18mobsacB derivative containing GMC with appropriate restriction enzymes, the fragments were ligated and transformed into *E. coli* S17-1 cells. The resulting *E. coli* S17-1 derivatives that carry pK18mobsacB containing GMC and the 5′ and 3′ regions of the target gene were mated with *P. aeruginosa* strains on LB plates at 37 °C for 48 h, and then the recipient *P. aeruginosa* cells with the first crossover were selected on LB plates containing chloramphenicol and gentamicin. After the occurrence of single crossover was confirmed by PCR, the second crossover was performed by culturing the single-crossover mutants on LB plates containing 12% sucrose. Double-crossover allelic exchange mutants were identified by PCR using the 5′ region upstream primer and the 3′ region downstream primer. For overexpression or complementation in *P. aeruginosa*, the pME6032 derivatives were transformed into relevant strains and induced by addition of 0.5 mM isopropyl β-D-1-thiogalactopyranoside (IPTG).

The deletion of *luxS* in *E. coli* strain BL21(DE3) was performed using the CRISPR-Cas9 system[44]. In brief, the sequence of single guide RNA (sgRNA) that contains a 20-bp guide sequence complementary to the target site within the *luxS* gene was amplified by PCR from plasmid pTargetF1 using the primer pair Δ*luxS*-sg20-F (containing the 20-bp guide sequence) and Δ*luxS*-sg20-R (Supplementary Table 2). The sgRNA fragment was ligated together with the 5′ and 3′ regions of the *luxS* gene amplified from genomic DNA by overlap extension PCR. The resulting PCR products were inserted into the SpeI/SalI sites of pTargetF1. The pTargetF1 derivative containing the sgRNA sequence and the 5′ and 3′ regions of the *luxS* gene were electroporated into *E. coli* BL21(DE3) competent cells harboring plasmid pCas in which arabinose (10 mM final concentration) have been added for induction of λ-Red recombinase[44]. Cells were recovered at 30 °C for 1 h and spread onto LB agar containing chloramphenicol and kanamycin. *E. coli* BL21(DE3) mutants with deletion of the *luxS* gene were identified by PCR and DNA sequencing, and then the pTargetF1 derivative and pCas in the Δ*luxS* mutant were successively eliminated by IPTG induction and overnight incubation at 37 °C, respectively[44].

**Chemotaxis assays**. For soft agar plate gradient assays, *P. aeruginosa* strains were grown overnight in M9 minimal medium[45] supplemented with 0.1% (w/v) glucose, washed twice with fresh M9 medium, and diluted to an $OD_{600}$ of 0.6 in M9 medium. 10 µl aliquots of 100 µM DPD/AI-2 (Omm Scientific) were placed onto M9 plates containing 2.5 mM glucose and 0.25% (w/v) Bacto agar. After overnight incubation at 4 °C for gradient formation, 2 µl aliquots of bacterial suspensions in M9 medium were placed horizontally to each of the DPD/AI-2 spots. All plates were incubated at 30 °C for 20 h and then examined for chemotaxis. The distance from the site of inoculation to the colony edges closest to (D1) and furthest from (D2) the DPD/AI-2 source was measured and the response index (RI) values were calculated as follows: $RI = D1/(D1 + D2)$. Colonies with RI values larger than 0.52 were considered to indicate chemotaxis[23].

For quantitative capillary chemotaxis assays, overnight cultures of *P. aeruginosa* strains in LB medium were diluted to an $OD_{600}$ of 0.05 in MS medium[45] supplemented with 15 mM glucose, 6 mg l$^{-1}$ Fe citrate and trace elements, and then grown at 37 °C until the $OD_{600}$ reached 0.4. After centrifugation, the pellet was washed twice with chemotaxis buffer (50 mM potassium phosphate, 20 mM EDTA, 0.05% glycerol, pH 7.0) and resuspended in the same buffer at an $OD_{600}$ of 0.1. Subsequently, 230 µl aliquots of bacterial suspensions were placed into the wells of a 96-well plate. Capillaries (Sigma cat# P1424) were heat sealed at one end, filled with chemotaxis buffer or DPD/AI-2 solutions dissolved in chemotaxis buffer, and then immersed into bacterial suspensions at their open ends. After incubation for 30 min at room temperature, the capillaries were removed and rinsed with sterile water. The sealed ends of the capillaries were broken and their contents were emptied into 1 ml of M9 medium. Serial dilutions were plated on LB agar plates, and the CFU were determined after incubation at 37 °C for 24 h. In all cases, cell numbers were corrected by subtracting the number of cells that swam into the buffer-containing capillaries.

**In vitro AI-2 binding assays**. Derivatives of pET-28a carrying the DNA fragments encoding the LBDs of signal transduction proteins were transformed into *E. coli* strain BL21(DE3) or its derivative lacking *luxS*. The resulting strains were grown at 37 °C in LB medium to an $OD_{600}$ of 0.8, shifted to 20 °C and induced with 0.25 mM IPTG for 7 h before harvest. After pellets were resuspended, cells were disrupted by sonication and then purified with Ni-nitrilotriacetic acid (Ni$^{2+}$-NTA) His-binding resin (Novagen, Madison, WI) according to the manufacturer's instructions. The proteins were eluted from the column using a solution containing 50 mM NaH$_2$PO$_4$ (pH 8.0), 300 mM NaCl and 250 mM imidazole, and then swapped into 50 mM NaH$_2$PO$_4$ (pH 8.0), 300 mM NaCl, and 1 mM dithiothreitol (DTT) using Sephadex-G25 agarose. After verifying the purity by SDS-PAGE analysis (Supplementary Fig. 13), purified proteins were concentrated to ~10 mg ml$^{-1}$ and denatured by heating at 70 °C for 10 min to release any bound ligands. The denatured proteins were pelleted and the resulting supernatants were then tested for the presence or absence of AI-2 in the luminescence assays. For this assay, an overnight culture of *V. harveyi* MM32 grown in AB medium were diluted 1:5000 into fresh AB medium, and 90 µl aliquots of the diluted cells were added to 96-well microtiter plates (Corning cat# 3603). Subsequently, 10 µl aliquots of the supernatants from denatured proteins or a buffer control were added to the wells and the microtiter plates were incubated at 30 °C for 10 h with shaking at 170 r.p.m. Bioluminescence (counts per second) was measured using microplate reader Victor X3 (PerkinElmer, Waltham, MA, USA) and AI-2 activity is reported as fold induction relative to the light production induced by the buffer control.

**Isothermal titration calorimetry**. ITC experiments were performed at 20 °C using a Nano ITC Standard Volume isothermal calorimeter (TA Instruments, New Castle, DE). For all His$_6$-tagged recombinant proteins, the N-terminal His$_6$ tag was cleaved by His$_6$-tagged tobacco etch virus (TEV) protease, and a second round of Ni$^{2+}$-NTA affinity chromatography was performed to remove the TEV protease, the cleaved tag and any uncut fusion protein. The protein purity was examined using SDS-PAGE (Supplementary Fig. 14). The tag-free proteins were dialyzed against a Tris buffer (25 mM Tris, 150 mM NaCl, pH 7.5) and DPD/AI-2 (Omm

Scientific) was dissolved in the same buffer. After being degassed, 1 ml of the protein (70 µM) and 250 µl of the DPD/AI-2 solution (700 µM) were added to the sample cell and the syringe, respectively. The stirring speed was 200 r.p.m. and 25 injections were used each experiment. Three independent experiments were performed for each sample. In control experiments, the DPD/AI-2 solution (700 µM) was titrated into the buffer in the sample cell to obtain the heat of dilution (Supplementary Fig. 15a). ITC data were analyzed and fit with a one-site independent binding model using the NanoAnalyze software version 3.4 provided by the manufacturer, with the heat of dilution subtracted from the experimental titrations before data analysis.

**Biofilm formation assays**. For crystal violet quantification of biofilm formation, overnight cultures of relevant strains of *P. aeruginosa* were diluted with TSB medium to an $OD_{600}$ of 0.05, and 190 µl aliquots of the diluted cells were inoculated into each well of a 96-well microtiter plate (cat# 220400, Zhejiang Sorfa Life Science Research Co., Ltd., China). 10 µl aliquots of the DPD/AI-2 solution (2 µM) or a buffer control were added to the wells containing the diluted cells. After incubation at 37 °C for 48 h without shaking, culture supernatant was removed and the wells were washed twice with phosphate-buffered saline (PBS). Cells adhering to the wells were stained with 0.1% (w/v) crystal violet for 15 min and then washed three times with PBS. The bacteria-bound dye was dissolved in 200 µl of 95% ethanol and the absorbance was determined at 590 nm.

For confocal laser-scanning microscopy and image analysis of static biofilms, overnight cultures of *P. aeruginosa* strains carrying pME6032-mCherry were diluted with TSB medium to an $OD_{600}$ of 0.01 (~$2 \times 10^7$ CFU ml$^{-1}$). 190 µl aliquots of the diluted cells and 10 µl aliquots of the DPD/AI-2 solution (2 µM) or a buffer control were inoculated into confocal dishes (cat# BDD011035, Guangzhou Jet Bio-Filtration Co., Ltd., China). The cultures were incubated at 37 °C under static conditions and biofilms were visualized using a Revolution XD laser-scanning confocal microscope (Andor, Belfast, Northern Ireland) after 1, 10, and 19 h. The excitation/emission wavelength for mCherry was 568/590 nm. Images were reconstructed using the Imaris 9.0 software package (Bitplane, AG) and the biovolumes were calculated using COMSTAT 2.1[46] plugin in ImageJ software (version 1.48). Biovolumes were measured and calculated from three biological replicates and each biological replicate was derived from an average of five confocal images.

**In vitro reaction of SAH with Pfs and LuxS and ITC analysis in a borate-depleted system**. *E. coli* XL1-Blue cells that carry the pGEX-6P-1 derivatives containing *pfs* or *luxS* gene from *E. coli* BL21(DE3) were grown at 37 °C in LB medium to an $OD_{600}$ of 0.6, and then induced with 0.5 mM IPTG at 24 °C for 10 h. Cells were harvested and lysed by sonication, and then GST-tagged fusion proteins were purified with GST-binding resin (Novagen, Madison, WI) according to the manufacturer's instructions[47]. For boron removal, water, the SAH solution and Tris buffer were filtered through a borate anion-specific resin Amberlite IRA-743 (Sigma cat# 216445)[48]. In brief, 30 ml of Amberlite resin is used to remove boron from 1000 ml of solution in a 50-ml polypropylene column (cat# HC-0650-10, Beijing Ruida Henghui Science & Technology Development Co., Ltd., China) with the following steps: 150 ml of 3 M NH$_4$OH, 600 ml of distilled water, 300 ml of 1 M HCl, 150 ml of distilled water, 300 ml of 0.16 M HNO$_3$, 600 ml of distilled water followed by 1000 ml of the solution. The purified Pfs and LuxS proteins were dialyzed against the borate-depleted Tirs buffer (pH 7.5). In vitro DPD/AI-2 synthesis reaction was carried out for 1 h at 37 °C. The reaction mixtures contained 1 mg ml$^{-1}$ of the purified Pfs and LuxS proteins, 1 mM SAH and 25 mM Tirs buffer (pH 7.5). After incubation, the reaction mixtures were filtered through Amicon Ultra-4 filters (limited 3000-molecular-weight cutoff) (Millipore) to remove proteins from the reaction products. Only plastic supplies were used for all experiments involving borate-depleted reagents. The levels of DPD/AI-2 in the reaction products were estimated by detecting the yield of homocysteine using LC-MS/MS (AB SCIEX Triple Quad 6500+ LC-MS/MS System)[6,47]. To test the effect of boron on the interactions of AI-2 with PctA-LBD and TlpQ-LBD, 250 µl (25 injections, 10 µl per injection) of the boron-free products from reaction of SAH with Pfs and LuxS (concentrations of DPD/AI-2 in the products were estimated to be 13 µM) or the boron-free products supplemented with 150 µM boric acid were injected into 1 ml of the tag-free proteins (1.3 µM) which have been dialyzed against and diluted in the borate-depleted Tris buffer or the borate-depleted proteins supplemented with 150 µM boric acid in sample cells in ITC experiments. The heats of dilution for the reaction products with or without added 150 µM boric acid titrated into the borate-depleted buffer with or without added 150 µM boric acid in sample cells were subtracted from raw titration data in the final analysis (Supplementary Fig. 15b, c).

**Molecular docking analysis**. The crystal structure of PctA-LBD (PDB ID: 5T7M)[26] and (PDB ID: 6FU4)[24] retrieved from the Protein Data Bank were refined and optimized using the Protein Preparation Wizard tool[49] integrated in the Schrödinger suite (Schrödinger Release 2018-4, Schrödinger, LLC, New York, NY, 2018). The two AI-2 molecules *S*-THMF-borate and *R*-THMF and histamine were extracted from the crystal structures of the *S*-THMF-borate-LuxP (PDB ID: 1JX6)[6], *R*-THMF-LsrB (PDB ID: 1TJY)[7] and TlpQ-LBD-histamine (PDB ID: 6FU4)[24] complexes,

respectively, and further optimized by LigPrep (LigPrep, version 2.5, Schrödinger, LLC, New York, NY, 2011). The prepared ligands, which were allowed to be flexible, were docked into the proteins using the XP docking mode of the Glide program (version 8.1)[33] in Schrödinger. The best binding mode of each ligand was selected based on the lowest Glide XP docking score. The three-dimensional figures were displayed using Chimera version 1.13[50].

**Preparation of membrane fractions containing full-length PctA, KinD and rpHK1S-Z16.** The *pctA* gene was cloned with a C-terminal His$_6$ tag into pHSe5 and expressed in *E. coli* strain HCB721[22,32] that is defective in all known *E. coli* MCPs and cytoplasmic chemotaxis proteins except for the phosphatase CheZ. The genes encoding full-length KinD and rpHK1S-Z16 were cloned with a C-terminal His$_6$ tag into pHSe5 and expressed in the Δ*luxS* mutant of *E. coli* BL21(DE3). Cultures of these three strains were grown at 37 °C in LB medium to an OD$_{600}$ of 0.8, and then 0.5 mM IPTG was added to induce protein expression at 24 °C for 10 h before harvest. After cells were resuspended in a high-salt buffer (20 mM Na$_3$PO$_4$, pH 7.0; 2 M KCl; 10% glycerol; 5 mM EDTA; 5 mM DTT; 1 mM phenylmethanesulfonyl fluoride) and lysed by sonication, the membrane fractions containing full-length PctA, KinD or rpHK1S-Z16 were collected by four rounds (1 h per round) of ultracentrifugation at 200,000 × *g* and 4 °C using Optima ultracentrifuge XPN-100 and rotor 70 Ti (Beckman Coulter, USA). After each round of ultracentrifugation, the membrane fractions in pellets were resuspended in the high-salt buffer[51]. The inverted membrane vesicles of full-length PctA, KinD and rpHK1S-Z16 prepared by ultracentrifugation were further purified by Ni$^{2+}$-NTA affinity chromatography. Purified membrane fractions were dialyzed in a storage buffer (25 mM Tris, 150 mM NaCl, pH 7.5; 10% glycerol) and subjected to SDS-PAGE analysis to examine the purity (Supplementary Fig. 16). Protein concentration was measured using the Bradford method.

**In vitro methylation assays.** Hundred micrograms of membrane fractions containing PctA and 15 μg of His$_6$-tagged CheR1 were preincubated in a 100-μl reaction system containing 50 mM NaH$_2$PO$_4$ (pH 8.0) and 300 mM NaCl for 10 min at 30 °C. The reaction was initiated by adding 10 μM SAM and the reaction was allowed to proceed for another 1 h at 30 °C. The effect of AI-2 on PctA methylation was examined by incubating DPD/AI-2 (100 and 500 μM) in the reaction mixture accordingly. A control experiment was performed with no addition of SAM to check the initial methylation state of PctA. The reaction was stopped by adding 2× SDS-PAGE loading buffer. The reaction products were resolved by SDS-PAGE and stained with Coomassie brilliant blue. The gel bands corresponding to PctA were excised and used for detection and quantification of methylation by nano-LC-MS/MS.

**In vitro kinase assays.** Twenty micrograms of membrane fractions containing KinD was preincubated in a 50-μl reaction system containing 50 mM Tris-HCl (pH 7.8), 2 mM DTT, 25 mM NaCl, 25 mM KCl, 5 mM MgCl$_2$ for 10 min at 30 °C. The reaction was initiated by adding 100 μM ATP and DPD/AI-2 (0, 10, and 50 μM) was added to the mixture simultaneously with ATP. A control experiment was performed without ATP to check the initial phosphorylation state of KinD. After 30 min incubation at 30 °C, products were resolved by SDS-PAGE, and the gel bands corresponding to KinD were excised and used for detection and quantification of phosphorylation by nano-LC-MS/MS.

**In-gel tryptic digestion and nano-LC-MS/MS analysis.** Gel slices excised from SDS-PAGE gels were destained with three washes in 100 μl of 100 mM ammonium bicarbonate in 50% acetonitrile at 37 °C for 15 min. Destained gel pieces were dried in a SpeedVac (Thermo Fisher Scientific) for 15 min. 10 mM DTT in 25 mM ammonium bicarbonate was added to cover the gel pieces and incubated at 56 °C for 1 h. After cooling to room temperature, the supernatant was replaced by 55 mM iodoacetamide in 25 mM ammonium bicarbonate followed by 1 h incubation at room temperature in darkness. Gel slices were washed with 100 μl of 25 mM ammonium bicarbonate for 10 min, and then shrunk in acetonitrile for 10 min. Acetonitrile was removed and the gel slices were dried in a SpeedVac. The dried gel slices were swollen with 20 μl of 12.5 ng μl$^{-1}$ Pierce™ trypsin protease (MS-grade; Thermo Fisher Scientific) in 50 mM ammonium bicarbonate at 4 °C for 1 h. An additional 30 μl of 50 mM ammonium bicarbonate was supplemented, followed by overnight incubation at 37 °C. After digestion, supernatants were transferred to a new tube, and remaining peptides were extracted from the gel slices with 50 μl of 0.1% trifluoroacetic acid at 37 °C for 30 min. Combined extracts for each gel slice were dried in a SpeedVac. The dried peptide samples were resolubilized in 0.1% formic acid and analyzed in C18 reversed-phase column connected to an EASY-nLC 1000 interfaced via a Nanospray Flex ion source to an Orbitrap Fusion Tribrid mass spectrometer (Thermo Fisher Scientific). The LC mobile phase consisted of 0.1% formic acid (v/v) in water (A) and 0.1% formic acid in acetonitrile (B). The flow rate was 0.4 μl min$^{-1}$ and the gradient program was set as follows: 3-8% B for 5 min, 8-20% B for 40 min, 20–35% B for 10 min, 35–80% B for 3 min, and finally 80% B for 2 min. A data-dependent Top 20 method was used with precursor MS1 scan (*m/z* 350–1550) acquired in the Orbitrap at a resolution of 120,000, followed by Orbitrap HCD-MS/MS and OTHCD-MS/MS of the 20 most abundant multiply charged

precursors in the MS1 spectrum. MS2 spectra were acquired at a resolution of 30,000. Methylation on glutamic acid residues of PctA and phosphorylation on histidine residues of KinD were set as variable modifications. Protein identification and quantification were performed using Mascot Daemon version 2.5.1 (Matrix Science, Boston, USA) considering the specific modifications.

**In vitro DGC activity assays.** The DGC activity of rpHK1S-Z16 was determined by measuring the synthesis of c-di-GMP using an HPLC-based method[52]. Seventy micrograms of membrane fractions containing rpHK1S-Z16 was added to a 200-μl reaction system containing 50 mM Tris-HCl (pH 7.5) and 5 mM MgCl$_2$ with 0, 100 or 200 μM DPD/AI-2. The reaction was initiated by adding 100 μM GTP and the reaction was allowed to proceed for 0, 30 and 60 min at 30 °C. At indicated time points, 50 μl aliquots were removed and heated at 100 °C for 5 min. Denatured proteins were removed through centrifugation and the supernatants were filtered through a 0.22 μm membrane. Samples were injected into an HPLC (Agilent 1260 infinity II) system equipped with a C18 reversed-phase column and a UV detector. Components were eluted isocratically with 98% A (150 mM Na$_2$HPO$_4$, pH 5.2) and 2% B (acetonitrile) in 15 min at a flow rate of 1 ml min$^{-1}$. The detection wavelength was 252 nm. GTP (Sigma, Cat# G8877) and c-di-GMP (Sigma, Cat# SML1228) were run as standards. The levels of synthesized c-di-GMP in the supernatants were determined from the standard curve obtained using known concentrations of c-di-GMP.

**Identification of dCache_1-containing proteins.** The amino acid sequences of 18970 dCache_1-containing proteins were downloaded from the Pfam 32.0 database (http://pfam.xfam.org/) based on Uniprot 2018_04 release. Domain predictions were carried out with PfamScan[53] at E-value threshold of 1E-5. Uniprot accessions, Pfam domain architectures, output domains of the dCache_1-containing proteins and their taxonomy are listed in Supplementary Data 1.

**Illustration of conserved residues using WebLogo 3.** 18970 dCache_1 sequences were aligned using ClustalW embedded in MEGA7 software[54] (Supplementary Data 3). Aligned columns not represented in the dCache_1 domain of PctA (residues 36-261) were removed and conserved residues were illustrated using the WebLogo 3 server (http://weblogo.threeplusone.com/). Amino acid numbering was based on the sequence of PctA.

**Statistical analysis.** All experiments were repeated at least three times with similar results. Data were statistically analyzed with GraphPad Prism 7.0 (GraphPad Software Inc), using two-sided, unpaired Student's *t*-test. Data are presented as mean ± s.d. or s.e.m. Differences were considered statistically significant at *p* < 0.05.

**Reporting summary.** Further information on research design is available in the Nature Research Reporting Summary linked to this article.

## Data availability

The protein sequence and domain data are available from the Pfam database and Uniprot database. Protein 3D coordinate data are available from the Protein Data Bank (http://www.rcsb.org). All the other data that support the findings of this study are available within the paper and its Supplementary Information and Supplementary Data or from the corresponding authors upon reasonable request. Source data are provided with this paper.

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

## Acknowledgements
We thank Dr. Sheng Yang at Institute of Plant Physiology and Ecology, Chinese Academy of Sciences for providing the CRISPR-Cas9 System, and Dr. Zhao-Xun Liang at Nanyang Technological University for providing *E. coli* strain HCB721 and the plasmid pHSe5. This work was supported by grants from the National Key R&D Program of China 2018YFA0901200 (to X.S. and L.Z.), the National Natural Science Foundation of China 31770121 (to L.Z.) and 31725003 (to X.S.), and the Fundamental Research Fund for the Central Universities 2452020181 (to L.Z.). We thank Dr. Jingfang Liu (Public Technology Service Center Institute of Microbiology, Chinese Academy of Sciences) for her help in identification of methylation and phosphorylation sites with mass spectrometry, and Dr. Zeyong Chen for molecular docking analysis. We also thank the Teaching and Research Core Facility at College of Life Science (Min Duan and Ningjuan Fan) and Life Science Research Core Services, NWAFU (Luqi Li) for technical support.

## Author contributions
L.Z. and X.S. conceived the ideas and designed the experiments; Unless otherwise specified, L.Z. and S.L. performed all of the experiments. L.Z., S.L., and Z.Wang performed the chemotaxis assays; S.L., X.L., and Z.Wei performed the gene deletions. S.L., M.J., R.W., L.X., X.X., Q.L., D.S., Z. Wang, and M.L. performed protein expression and purification experiments. C.F., L.Z., and S.L. participated in the molecular docking analysis. L.Z., S.L., and X.L. performed the computational analyses. L.Z., Z.Q.L., Y.W., and X.S. analyzed data. L.Z., X.S., and Z.Q.L. wrote the paper. All authors discussed the results and commented on the manuscript.

## Competing interests
The authors declare no competing interests.
