## [Peer Review File · Nature Communications]

REVIEWER COMMENTS

Reviewer #1 (Remarks to the Author):

The authors show that *P. aeruginosa* performs chemotaxis to autoinducer 2 and the analysis of mutants in chemoreceptor genes indicates that the chemoreceptors PctA and TlpQ are responsible for AI-2 chemotaxis. In addition, AI-2 mediated biofilm formation in a PctA and TlpQ dependent manner. Microcalorimetric studies then revealed that AI-2 bound to the individual ligand binding domains of PctA and TlpQ. Evidence was obtained for there being other dCACHE domains in chemoreceptors and other bacterial signal transduction systems that equally bind AI-2. These are surprising results since PctA and TlpQ have previously been identified as chemoreceptors that bind specifically amino acids or polyamines/histamine. This is an interesting study but this referee has major concerns on a series of experimental issues.

ITC experiments: The authors report K_D and n -values derived from microcalorimetric titrations. The n -value represents the ligand:protein stoichiometry. In all cases n -values close to 1 were reported indicating that one ligand binds to one protein molecule. However, the n -value can be estimated from sigmoidal titration curves and corresponds to the molar ratio (lower X-axis) at the point of inflection of the binding curve. In all experiments there were enormous differences between the visually estimated n -value and the n -value presented by the authors. For example, in Figs. 2b and c this referee estimates an n -value of approximately 10, whereas values of approximately 1 are presented by the authors. The estimated stoichiometry would thus be 10 molecules of AI-2 binding to one protein molecule.

Microcalorimetric data need to be corrected using titrations of buffer with the ligand. However, data have not been corrected and no control experiment is shown. This is even more important since the ligand injected is not a homogeneous solution of stable ligand but a mixture of different, interchangeable forms of DPD/AI-2. It cannot be excluded that upon dilution (injection) the ratio of the different interchangeable forms are altered and that the corresponding heat changes are registered.

In the materials and methods section the authors state that 10 micromolar protein were titrated with aliquots of a 700 micromolar AI-2. However, there is evidence that a number of different protocols have been used and the corresponding experimental detail is not provided. As an example this referee cites Fig. 2b and Extended data Fig. 2b, representing the titration of PctA-LBD and PctC-LBD with AI-2. The authors have shown that deletion of the *pctA* reduces AI-2 chemotaxis, whereas deletion of *pctC* had no effect.

In the upper panels (titration raw data) the initial peak was of approx. 35 microJ/sec for PctA, whereas in the case of PctC the initial peak was of only 0.9 microcal/sec, corresponding to approx. 3.5 microJ/sec (note: the use of both, J and cal in the titrations is highly confusing). However, when these raw data are normalized with the ligand concentration used (lower panel) the initial injection peak corresponded to 95 kJ/mol of DPD/AI-2 for PctA, but to 110 kJ/mol of DPD/AI-2 for PctC.

This indicates that PctA-LBD was titrated with a 15 fold higher DPD/AI-2 concentration than PctC-LBD. This notion is also supported by the molar DPD/AI-2:protein ratio ratio at the end of the titration, that was approximately 45 in the case of PctA-LBD and only 3 for PctC-LBD.

Hence the question: Why were PctA-LBD and TlpQ-LBD titrated with a significantly higher DPD/AI-2 concentration than PctB-LBD and PctC-LBD? Increasing ligand concentration will favor complex formation. Although the authors state that AI-II did not bind to PctC-LBD, they report in Extended data Fig. 2 a K_D of 35 micromolar. This is an affinity with which many chemoreceptors recognize physiologically relevant ligands.

Almost, no information is presented as to the knowledge available on the PctA, PctB, PctC and TlpQ chemoreceptors. Both, PctA and TlpQ LBD had been submitted to high throughput ligand screening experiments with commercially available ligand collections that revealed that PctA-LBD binds specifically amino acids (McKellar et al. Mol. Micro) and TlpQ-LBD binds specifically histamine and polyamines (Corral-Lugo et al. (2018) mBio). Both 3D structures have been determined and the molecular basis for ligand specificity has been determined. These data do not mean that both

receptors may not bind AI-2, but the information needs to be provided to provide the correct context of the data presented by the authors. In addition PctA has two paralogs, PctB and PctC, that share a high degree of sequence identity and that also bind amino acids. Five highly conserved residues were detected in the binding site that corresponded to R126, W128, Y144, D146 and D173 and the authors suggest that these residues form a motif specific for AI-2 binding CACHE domains. However, in a recent publication (Gavira et al. (2020) mBio 11, e03066-19) the structure of PctA-LBD was reported. If the authors inspect Fig. 6A of this publication, they will see that all the five amino acids cited form interactions with the bound amino acid. Based on these observations Gavira et al defined a sequence motif that was proposed to identify amino acid binding Cache domains. There is extensive experimental evidence that in fact this motif occurs in amino acid binding dCACHE domains. The following dCache domains were found to contain this motif and experimental evidence was presented that indeed the corresponding receptor binds amino acids: CtaA, CtaB and CtaC (*P. fluorescens*) Oku et al. (2012), *Microbes Environ.* 27:462-9, Oku et al. (2014) *Microbes Environ.* 29:413-9); PscA and PscB (*P. syringae*) McKellar et al. (2015) *Mol Microbiol.*96:694-707 and Cerna-Vargas et al. (2019) mBio. 10(5). pii: e01868-19); McpG (*P. putida*) Reyes-Darias et al. (2015) *Mol Microbiol.* 97:488-501) and McpA (*P. putida*) Corral-Lugo et al. (2016) *Environ Microbiol.* 18:3355-3372.

In addition this motif is not conserved in the 3D Structure of TlpQ (pdb ID: 6Fu4, Corral-Lugo et al. (2018) mBio 13: e01894-18). Instead the authors of this publication identify in the binding site of TlpQ a sequence motif that is present in other dCache domains that bind compounds with amino groups. Taken together, there is strong evidence that the five amino acid motif identified is characteristic for the dCache subfamily that recognizes amino acids.

The authors conducted docking experiments with AI-2 to the membrane distal module of PctA-LBD. Fig. 4D shows the ligand position with the lowest docking score. However, what was the docking score and how did it compare to the docking scores of different amino acids to the membrane distal module as reported in Gavira et al. (2020) mBio 11, e03066-19? The structure of TlpQ-LBD is also available. Did the authors conduct similar experiments with this structure?

Minor:

Line 98: "Whereas most of these mutants did not affect the response of *P. aeruginosa* to AI-2..", to be rephrased

Line 110:

Line 110: "HHpred analysis suggests that their structures are highly similar (E-value=4.7e-22111)." This sentence is ambiguous. HHpred is a fast server for remote protein homology detection and structure prediction. However, high resolution 3D structures of PctA-LBD and TlpQ-LBD have been published and the coordinates are available at the protein data bank; there is no structure prediction necessary.

Reviewer #2 (Remarks to the Author):

Decades ago, the quorum-sensing autoinducer-2 (AI-2) and its synthase LuxS were discovered in vibrios. Production of this autoinducer was shown to be widespread in the bacterial domain, leading to the idea that AI-2 likely functions as a "universal" interspecies quorum-sensing signal. Despite this notion, only two AI-2-binding receptors have been discovered to date, LuxP, found exclusively in vibrios, and LsrB from enteric bacteria and limited other families, calling into question the "universality" of AI-2 signaling. Numerous bacteria that do not possess homologues of the AI-2 receptors have been shown to respond to AI-2 as measured by phenotypic and gene expression studies, leading to the hypothesis that other, undiscovered, AI-2 receptors must exist. In the current study, Zhang et al. set out to determine how the opportunistic pathogen, *Pseudomonas aeruginosa*, which lacks LuxP and LsrB homologues, senses and responds to AI-2. In the process, the authors identify a novel AI-2 binding domain that is widely present in proteins in

diverse organisms, and moreover, they show that AI-2 binding by these proteins drives changes in downstream traits. This discovery provides significant new and convincing evidence that AI-2 is indeed a signaling molecule that is used across the bacterial world.

The authors first show that *P. aeruginosa* exhibits chemotaxis to AI-2, which has previously been demonstrated in other organisms. Given this result, the authors reasoned that one or more of the 26 chemoreceptors in *P. aeruginosa* must detect AI-2. Evaluating the chemotactic responses of mutants lacking each individual chemoreceptor revealed PctA and TlpQ to be required for AI-2 chemotaxis. The authors thoroughly demonstrate, using reporter assays and isothermal titration calorimetry, that the ligand binding domains (LBD) of both chemoreceptors preferentially bind the non-borated form of AI-2 with nanomolar affinity, proving that these proteins are indeed novel AI-2 receptors. They go on to identify residues within the LBD of PctA that are critical for AI-2 binding.

To explore the conservation of the new AI-2 binding module, the authors conducted a bioinformatic search for proteins possessing a domain similar to the PctA LBD. This strategy revealed many such proteins from divergent organisms, including other chemoreceptors, kinases, and diguanylate cyclases. The authors convincingly demonstrate AI-2 binding to a subset of these proteins and they show the enzymatic activities of one histidine kinase and one diguanylate cyclase are regulated by AI-2. Finally, the authors identify ~1500 proteins from diverse organisms (including archaea) that are potential AI-2 binding receptors. Notably, these proteins, all of which have dCACHE_1 domains, are predicted to have a variety of activities that could be regulated by AI-2. In a lovely finale, the authors randomly select 19 of these dCACHE_1 proteins and demonstrate AI-2 binding to two of them using a reporter assay.

Overall, this manuscript is thorough and interesting and it provides a significant new discovery. The experiments are well controlled, and the manuscript is written in a fairly straightforward manner.

One major problem with the manuscript is that there are numerous instances of plagiarism. Hopefully, this is an oversight on the part of the authors. Whole passages, with only insignificant changes from several of the foundational AI-2 studies have been lifted and patched together in the introduction and the discussion sections. The authors have also included an exact replica of a figure panel from the original study that defined the AI-2 biosynthetic pathway without credit. This manuscript cannot be considered until these plagiarism issues, and several other points, outlined below, are addressed.

Major concerns

1. There are multiple incidences of plagiarism. Lines 41-56 of the introduction bear such uncanny resemblance to passages from the original AI-2 papers that the line between describing what has come before and plagiarism has been crossed. The introduction needs to be rewritten by the authors and they need to change more than a word here or there or modestly change the order of the words/sentences from the original references. While not as egregious, the same goes for the initial parts of the discussion. Phrases like "monitor ligand occupancy" etc. come straight from earlier papers. If these authors want to review other scientists' work, they need to be careful not to lift language, but rather, use their own wording. Furthermore, regarding Fig. 1A, they need to give a reference for the biosynthesis scheme. That exact figure comes from someone else's published work and it needs to be credited, otherwise it is plagiarism. The authors also need to include the original reference (Chen) showing the AI-2 molecule makes the boron adduct. Chemotaxis to AI-2 has also previously been demonstrated in other organisms. The authors need to credit these ideas/studies earlier in the paper than in the discussion.

2. It is unclear what the co-culture biofilm results presented in Figure 3B-C contribute to the authors' message. These experiments are complex, challenging to interpret, and the results could be indirect. For example, the authors show that while WT *S. aureus* stimulates *P. aeruginosa* biofilm formation, the Δ luxS *S. aureus* does not. Does the *S. aureus* Δ luxS grow at the same rate

as WT in the presence of *P. aeruginosa*? What genes does AI-2 regulate in *S. aureus* that could affect the interaction between the two species? The authors claim that the result shows that *P. aeruginosa* is detecting AI-2 made by *S. aureus*. That claim is not supported by the data. Either the *P. aeruginosa* senses AI-2 made by *S. aureus*, or, equally possible is that AI-2 mediated changes alter the interactions in biofilms. This figure and companion text should be removed. The experiment does not fit well with the other points made, and most importantly, it weakens the paper. Unlike the other experiments, which are rigorous, this one is unconvincing.

3. There are no gel images or indications as to purity levels of proteins. Please provide SDS-PAGE gel images in a supplementary figure.

4. Why are the K_d 's so different in Figs 2 and 4? Also, within that context, the K_d in Fig 2 and in Fig 4D is $0.16 \mu\text{M}$, but in Fig 4A it is 56 nM with no boron. If there was boron present, the authors show that the K_d should be $3 \mu\text{M}$. Please reconcile.

Minor concerns

1. In Lines 212-220-what is the explanation for the proteins that did not bind AI-2? These are all close homologs, so why do the authors think two bind AI-2 and four others do not? This result needs some text explanation.

2. In the images in 3B, the results are difficult to see (particularly the abundance of *S. aureus*). As mentioned in point 2 above, probably this set of experiments should be removed altogether. At most, these images could go in the supplement as non-merged, side by side panels that would enable the reader to interpret the data. The bar plot in C is sufficient for the main text. But again, it weakens the strong message, so better to remove all of these data.

3. The lettering of figure panels is inconsistent. For example, in Fig 1, the lettering proceeds in the standard order (left to right, top to bottom) in Fig 2, a and d are next to each other on the top of the figure. While this is a tiny point, it will be a lot easier on the reader if the layouts are made to be consistent.

4. In the discussion, please speculate on the possibility of inter-kingdom signaling via AI-2 given the presence of dCACHE_1 domains in archaeal proteins. Furthermore, please remove "tempted to speculate". It is a cliché. Just speculate. It is the discussion, speculating is what a discussion is for, and moreover, it is what the authors are doing.

5. There are many tense and grammatical mistakes. As an example, Line 342, "medium till OD600 of 0.8," should be corrected to "medium to an OD600 of 0.8". "Casted" is not a word.... The manuscript needs to be thoroughly edited.

6. A 2000 page supplement is not a good idea. Perhaps the authors could provide a link to the search results and host it in a data repository.

Response to Reviewers

We wish to begin by thanking the two reviewers for their very supportive and constructive comments. Please find our detailed responses to each of the comments below.

Reviewer #1 (Remarks to the Author):

The authors show that *P. aeruginosa* performs chemotaxis to autoinducer 2 and the analysis of mutants in chemoreceptor genes indicates that the chemoreceptors PctA and TlpQ are responsible for AI-2 chemotaxis. In addition, AI-2 mediated biofilm formation in a PctA and TlpQ dependent manner. Microcalorimetric studies then revealed that AI-2 bound to the individual ligand binding domains of PctA and TlpQ. Evidence was obtained for there being other dCACHE domains in chemoreceptors and other bacterial signal transduction systems that equally bind AI-2. These are surprising results since PctA and TlpQ have previously been identified as chemoreceptors that bind specifically amino acids or polyamines/histamine. This is an interesting study but this referee has major concerns on a series of experimental issues.

Response: We would like to thank the reviewer for the very positive comments on our study.

ITC experiments: The authors report KD and n-values derived from microcalorimetric titrations. The n-value represents the ligand:protein stoichiometry. In all cases n-values close to 1 were reported indicating that one ligand binds to one protein molecule. However, the n-value can be estimated from sigmoidal titration curves and corresponds to the molar ratio (lower X-axis) at the point of inflection of the binding curve. In all experiments there were enormous differences between the visually estimated n-value and the n-value presented by the authors. For example, in Figs. 2b and c this referee estimates an n-value of approximately 10, whereas values of approximately 1 are presented by the authors. The estimated stoichiometry would thus be 10 molecules of AI-2 binding to one protein molecule.

Microcalorimetric data need to be corrected using titrations of buffer with the ligand. However, data have not been corrected and no control experiment is shown. This is even more important since the ligand injected is not a

homogeneous solution of stable ligand but a mixture of different, interchangeable forms of DPD/AI-2. It cannot be excluded that upon dilution (injection) the ratio of the different interchangeable forms are altered and that the corresponding heat changes are registered.

In the materials and methods section the authors state that 10 micromolar protein were titrated with aliquots of a 700 micromolar AI-2. However, there is evidence that a number of different protocols have been used and the corresponding experimental detail is not provided. As an example this referee cites Fig. 2b and Extended data Fig. 2b, representing the titration of PctA-LBD and PctC-LBD with AI-2. The authors have shown that deletion of the *pctA* reduces AI-2 chemotaxis, whereas deletion of *pctC* had no effect.

In the upper panels (titration raw data) the initial peak was of approx. 35 microJ/sec for PctA, whereas in the case of PctC the initial peak was of only 0.9 microcal/sec, corresponding to approx. 3.5 microJ/sec (note: the use of both, J and cal in the titrations is highly confusing). However, when these raw data are normalized with the ligand concentration used (lower panel) the initial injection peak corresponded to 95 kJ/mol of DPD/AI-2 for PctA, but to 110 kJ/mol of DPD/AI-2 for PctC.

This indicates that PctA-LBD was titrated with a 15 fold higher DPD/AI-2 concentration than PctC-LBD. This notion is also supported by the molar DPD/AI-2:protein ratio at the end of the titration, that was approximately 45 in the case of PctA-LBD and only 3 for PctC-LBD.

Hence the question: Why were PctA-LBD and TlpQ-LBD titrated with a significantly higher DPD/AI-2 concentration than PctB-LBD and PctC-LBD? Increasing ligand concentration will favor complex formation. Although the authors state that AI-II did not bind to PctC-LBD, they report in Extended data Fig. 2 a K_D of 35 micromolar. This is an affinity with which many chemoreceptors recognize physiologically relevant ligands.

Response: We thank the reviewer for the insightful comment. In the ITC experiments for DPD/AI-2 binding, 1 ml of 10 μ M protein were titrated with 250 μ l of 700 μ M DPD/AI-2 (25 injections, 10 μ l per injection). The K_d and binding stoichiometry (n) were calculated by the ITC Run software. However, when ITC data were analyzed and fit with a one-site independent binding model using the NanoAnalyze software, we did not carefully check the parameters,

and the incorrect initial titrate volume of 250 μl was used for PctA-LBD, PctB-LBD, TlpQ-LBD, KinD-LBD and rpHK1S-Z16-LBD. For PctC-LBD, when ITC data were fit with a one-site independent binding model by the NanoAnalyze software, the initial titrate volume is correct (1000 μl) but blank (constant) model was not selected. These mistakes led to the abnormal one-site binding curves. In the revised manuscript, we have proofed all the parameters and corrected the initial titrate volume to 1000 μl to get the one-site binding titration curves, and the K_d and binding stoichiometry (n) were also calculated by the NanoAnalyze software, thus ensuring consistency between the visually estimated n -value and the n -value presented by us. As suggested, in all ITC experiments, microcalorimetric data have been corrected using titrations of buffer with the ligands (700 μM DPD/AI-2, the products of the Pfs/LuxS reaction in a borate-depleted system, or the products of the Pfs/LuxS reaction in a borate-depleted system supplemented with 150 μM boric acid) and one representative of the control experiments is shown (**Supplementary Fig. 13**).

The binding affinity of PctB-LBD and PctC-LBD for AI-2 is approximately 30 μM . As suggested, we have revised the statement “no binding of AI-2 to PctB-LBD or PctC-LBD” to “low-affinity binding of AI-2 to PctB-LBD ($28.05 \pm 2.39 \mu\text{M}$) and PctC-LBD ($39.94 \pm 3.44 \mu\text{M}$)” (**Line 144**). Although this is an affinity with which many chemoreceptors recognize physiologically relevant ligands such as amino acids, previous studies have showed that in culture supernatants peak production of AI-2 could reach levels of only approximately 2.5 μM (Yu *et al.* Infect. Immun. 2013, 81: 4053-4062). Furthermore, our results showed that deletion of *pctB* or *pctC* did not affect the response of *P. aeruginosa* to AI-2 (**Fig. 1d**). Together, these results suggest that AI-2 is not a physiologically relevant ligand for PctB and PctC.

Almost, no information is presented as to the knowledge available on the PctA, PctB, PctC and TlpQ chemoreceptors. Both, PctA and TlpQ LBD had been submitted to high throughput ligand screening experiments with commercially available ligand collections that revealed that PctA-LBD binds specifically amino acids (McKellar *et al.* Mol. Micro) and TlpQ-LBD binds specifically

histamine and polyamines (Corral-Lugo et al. (2018) mBio). Both 3D structures have been determined and the molecular basis for ligand specificity has been determined. These data do not mean that both receptors may not bind AI-2, but the information needs to be provided to provide the correct context of the data presented by the authors. In addition PctA has two paralogs, PctB and PctC, that share a high degree of sequence identity and that also bind amino acids.

Response: We thank the reviewer for this very insightful point. As suggested, the information that PctA-LBD specifically binds amino acids (McKellar, *et al.* Mol. Microbiol. 2015, 96: 694-707) and TlpQ-LBD specifically binds histamine and polyamines (Corral-Lugo *et al.* mBio 2018, 9, e01894-18) and the 3D structures of PctA-LBD (PDB ID: 5LTX) and TlpQ-LBD (PDB ID: 6FU4) have been supplemented to provide the correct context of the data (**Lines 119-125**). Although PctA, PctB and PctC are paralogous chemoreceptors, they bind distinct amino acid ligands (Rico-Jiménez *et al.* Mol. Microbiol. 2013, 88: 1230–1243; Gavira *et al.* mBio 2020, 11: e03066-19). Thus, it is not surprising that PctA, but not PctB or PctC, shows high affinity for AI-2. Furthermore, PctB-LBD does not contain the conserved residue corresponding to D146 of PctA (E146 in PctB), nor does PctC-LBD contain the conserved residue corresponding to Y144 of PctA (F147 in PctC) (**Supplementary Fig. 12**). Mutations of the nonconserved residue to conserved residue within PctB-LBD (E146D) and PctC-LBD (F147Y) increased their AI-2 binding affinity to levels (0.24-0.28 μM) (**Supplementary Fig. 3**) comparable to that of PctA-LBD (0.18 μM) (**Fig. 2b**), indicating that the changed key residue in PctB-LBD (E146) and PctC-LBD PctC-LBD (F147) is the main cause of their low affinity with AI-2.

Five highly conserved residues were detected in the binding site that corresponded to R126, W128, Y144, D146 and D173 and the authors suggest that these residues form a motif specific for AI-2 binding CACHE domains. However, in a recent publication (Gavira et al. (2020) mBio 11, e03066-19) the structure of PctA-LBD was reported. If the authors inspect Fig. 6A of this publication, they will see that all the five amino acids cited form interactions with the bound amino acid. Based on these observations Gavira et al defined a sequence motif that was proposed to identify amino acid binding Cache

domains. There is extensive experimental evidence that in fact this motif occurs in amino acid binding dCACHE domains. The following dCache domains were found to contain this motif and experimental evidence was presented that indeed the corresponding receptor binds amino acids: CtaA, CtaB and CtaC (*P. fluorescens*) Oku et al. (2012), *Microbes Environ.* 27:462-9, Oku et al. (2014) *Microbes Environ.* 29:413-9); PscA and PscB (*P. syringae*) McKellar et al. (2015) *Mol Microbiol.*96:694-707 and Cerna-Vargas et al. (2019) *mBio.* 10(5). pii: e01868-19); McpG (*P. putida*) Reyes-Darias et al. (2015) *Mol Microbiol.* 97:488-501) and McpA (*P. putida*) Corral-Lugo et al. (2016) *Environ Microbiol.* 18:3355-3372.

Response: We thank the reviewer for this important point. Indeed, we have noticed that the five highly conserved residues R126, W128, Y144, D146 and D173 of PctA form interactions with the bound amino acids such as L-Met, L-Ile and L-Trp (Gavira *et al.* *mBio* 2020, 11: e03066-19). In fact, the structures of the three amino acid ligands L-Met, L-Ile and L-Trp are notably different. In particular, L-Trp contains the indole ring which are not present in L-Met and L-Ile. It seems that the five highly conserved residues always play an important role when PctA-LBD bind structurally differentiated ligands, which is consistent with our observations that these five conserved residues represent a motif specific for AI-2 binding. In the recent publication by Gavira *et al.* (*mBio* 2020, 11: e03066-19), the authors identified a highly conserved region (HCR) corresponding to 121-129 of PctA that is specific for amino acids (Fig. 7 of this publication), but did not identify the five highly conserved residues R126, W128, Y144, D146 and D173 as amino acid binding motif. As mentioned above, PctB-LBD does not contain the conserved residue corresponding to D146 of PctA (E146 in PctB), while PctC-LBD does not contain the conserved residue corresponding to Y144 of PctA (F147 in PctC) (**Supplementary Fig. 12**). We also observe that the amino acid-binding chemoreceptors such as CtaB and CtaC (*P. fluorescens*), PscA and PscB (*P. syringae*) and McpA (*P. putida*) do not contain the conserved residue corresponding to D146 of PctA (**Supplementary data 3**). Obviously, the dCache domains involved in binding amino acids do not necessarily contain all the five highly conserved residues corresponding to R126, W128, Y144, D146 and D173 of PctA. Furthermore, CtaA (*P. fluorescens*) and McpG (*P. putida*) contain all the five conserved

residues (**Supplementary data 2**) and our ITC experiments showed that CtaA-LBD and McpG-LBD binds AI-2 with high affinity (**Unpublished data shown below**), further supporting our suggestion that five highly conserved residues in the binding site corresponding to R126, W128, Y144, D146 and D173 of PctA represent a motif for AI-2 binding.

AI-2 shows high affinity to CtaA-LBD (a) and McpG-LBD (b)

In addition this motif is not conserved in the 3D Structure of TlpQ (pdb ID: 6Fu4, Corral-Lugo et al. (2018) mBio 13: e01894-18). Instead the authors of this publication identify in the binding site of TlpQ a sequence motif that is present in other dCache domains that bind compounds with amino groups. Taken together, there is strong evidence that the five amino acid motif identified is characteristic for the dCache subfamily that recognizes amino acids.

Response: As we have discussed above, the dCache domains responsible for amino acid binding do not necessarily contain all the five amino acid motif, especially the residues corresponding to Y144 and D146 of PctA (the conserved residue corresponding to D146 in PctA is absent in PctB, CtaB, CtaC, PscA, PscB and McpA, and PctC does not contain the conserved residue corresponding to Y144 in PctA). In fact, TlpQ-LBD also contains four conserved residues corresponding to W128, Y144, D146 and D173 of PctA (**Supplementary Fig. 12**). The docking conformation of the boron-free

R-THMF to TlpQ-LBD (**Supplementary Fig. 8a**) showed that 192W, Y208, D210 and D239, which correspond to W128, Y144, D146 and D173 of PctA, lie closest to the ligand *R*-THMF (<2.7 Å), and mutations of each of these residues drastically reduced the binding affinity of TlpQ-LBD for AI-2 (**Supplementary Fig. 8b**). However, the residue N190 of TlpQ that corresponds to R126 of PctA lies far away from the ligand (5.16 Å) and does not seem to directly interact with the bound ligand. The mutant N190A shows similar AI-2 binding affinity as the WT protein (**Unpublished data shown below**), supporting the notion that there is no interaction between N190 of TlpQ and AI-2. The study by Corral-Lugo *et al.* (mBio 2018, 9: e01894-18) also showed that N190 of TlpQ does not form interactions with the bound histamine. Thus, the pockets of PctA-LBD and TlpQ-LBD do not use the exact same set of residues to bind AI-2. Furthermore, we also found that the AI-2-binding LBDs of KinD and rPHK1S-Z16 possess none of the five conserved residues (**Supplementary Fig. 12**). Thus, although we suggest that dCache_1 domains with the five conserved amino acid motif has the capacity to bind AI-2, we also point out the existence of other, yet unrecognized, AI-2-binding motifs (**Line 260-263, 318-321**). Overall, the high-affinity binding of AI-2 to dCACHE domains such as PctA-LBD, TlpQ-LBD, KinD-LBD and rPHK1S-Z16-LBD can easily be repeated, while the molecular basis for AI-2 binding remains to be elucidated.

AI-2 shows high affinity to TlpQ-LBD_{N190A}

The authors conducted docking experiments with AI-2 to the membrane distal module of PctA-LBD. Fig. 4D shows the ligand position with the lowest docking score. However, what was the docking score and how did it compare to the docking scores of different amino acids to the membrane distal module as reported in Gavira et al. (2020) mBio 11, e03066-19? The structure of TlpQ-LBD is also available. Did the authors conduct similar experiments with this structure?

Response: We thank the reviewer for raising this important question. In order to compare the docking scores of AI-2 to the membrane distal module of PctA-LBD with those of the docking of different amino acids to PctA-LBD, we conducted the docking analysis again using the extra precision (XP) docking mode of the Glide program in Schrödinger as described by Gavira *et al.* (mBio 2020, 11: e03066-19). The best conformation of *R*-THMF has the lowest docking score of $-7.82 \text{ kcal mol}^{-1}$, which is higher than those (ranging from -13.74 to $-11.53 \text{ kcal mol}^{-1}$) obtained from the docking of L-Met, L-Ile or L-Trp to PctA-LBD by Gavira *et al.* (mBio 2020, 11: e03066-19) but is below the lower threshold value of -6 kcal mol^{-1} for significance proposed by Gavira *et al.* (2020) (**Lines 196-208**). We also repeated the docking of L-Met, L-Ile and L-Trp to PctA-LBD and the resulting XP scores (L-Met, -10.14 ; L-Ile, -11.36 ; L-Trp, -13.84) are almost equal to those reported by Gavira *et al.* (2020), indicating the comparability of our results with those reported by Gavira *et al.* (2020). However, the best conformation of *S*-THMF-borate bound in the active pocket of PctA-LBD has the lowest docking score of $-2.92 \text{ kcal mol}^{-1}$, which is distant from the threshold for significance and thus suggests a weak interaction between *S*-THMF-borate and PctA-LBD. As suggested, we also performed the docking of the two AI-2 forms to TlpQ-LBD (**Lines 208-216**). All binding conformations of *S*-THMF-borate showed positive docking scores, suggesting unfavorable binding of *S*-THMF-borate to TlpQ-LBD. In contrast, the best conformation of *R*-THMF bound in the active pocket of TlpQ-LBD has the lowest docking score of $-5.70 \text{ kcal mol}^{-1}$, which is comparable to that ($-6.58 \text{ kcal mol}^{-1}$) obtained from the docking of histamine to TlpQ-LBD by us (**Supplementary Fig. 8a**). Furthermore, mutations in 192W, Y208, D210 and D239, which lie closest to the ligand *R*-THMF, drastically reduced the binding

affinity of TlpQ-LBD for AI-2, supporting the binding of TlpQ-LBD with *R*-THMF (Supplementary Fig. 8b).

Minor:

Line 98: “Whereas most of these mutants did not affect the response of *P. aeruginosa* to AI-2..”, to be rephrased

Response: We have revised these sentences as follow: “Whereas most of these mutations did not affect the response of *P. aeruginosa* to AI-2...” (Lines 108-109).

Line 110: “HHpred analysis suggests that their structures are highly similar (E-value=4.7e-22111).” This sentence is ambiguous. HHpred is a fast server for remote protein homology detection and structure prediction. However, high resolution 3D structures of PctA-LBD and TlpQ-LBD have been published and the coordinates are available at the protein data bank; there is no structure prediction necessary.

Response: We thank the reviewer for this important point. As suggested, we have performed alignment of the 3D structures of PctA-LBD (PDB ID: 5LTX) (Gavira *et al.* mBio 2020, 11: e03066-19) and TlpQ-LBD (PDB ID: 6FU4) (Corral-Lugo *et al.* mBio 2018, 9: e01894-18) using TM-align server, which suggests that they are in about the same fold (TM-score=0.79, Supplementary Fig. 2) (Lines 120-122).

Reviewer #2 (Remarks to the Author):

Decades ago, the quorum-sensing autoinducer-2 (AI-2) and its synthase LuxS were discovered in vibrios. Production of this autoinducer was shown to be widespread in the bacterial domain, leading to the idea that AI-2 likely functions as a “universal” interspecies quorum-sensing signal. Despite this notion, only two AI-2-binding receptors have been discovered to date, LuxP, found exclusively in vibrios, and LsrB from enteric bacteria and limited other families, calling into question the “universality” of AI-2 signaling. Numerous bacteria that do not possess homologues of the AI-2 receptors have been shown to respond to AI-2 as measured by phenotypic and gene expression studies, leading to the hypothesis that other, undiscovered, AI-2 receptors

must exist. In the current study, Zhang et al. set out to determine how the opportunistic pathogen, *Pseudomonas aeruginosa*, which lacks LuxP and LsrB homologues, senses and responds to AI-2. In the process, the authors identify a novel AI-2 binding domain that is widely present in proteins in diverse organisms, and moreover, they show that AI-2 binding by these proteins drives changes in downstream traits. This discovery provides significant new and convincing evidence that AI-2 is indeed a signaling molecule that is used across the bacterial world.

The authors first show that *P. aeruginosa* exhibits chemotaxis to AI-2, which has previously been demonstrated in other organisms. Given this result, the authors reasoned that one or more of the 26 chemoreceptors in *P. aeruginosa* must detect AI-2. Evaluating the chemotactic responses of mutants lacking each individual chemoreceptor revealed PctA and TlpQ to be required for AI-2 chemotaxis. The authors thoroughly demonstrate, using reporter assays and isothermal titration calorimetry, that the ligand binding domains (LBD) of both chemoreceptors preferentially bind the non-borated form of AI-2 with nanomolar affinity, proving that these proteins are indeed novel AI-2 receptors. They go on to identify residues within the LBD of PctA that are critical for AI-2 binding.

To explore the conservation of the new AI-2 binding module, the authors conducted a bioinformatic search for proteins possessing a domain similar to the PctA LBD. This strategy revealed many such proteins from divergent organisms, including other chemoreceptors, kinases, and diguanylate cyclases. The authors convincingly demonstrate AI-2 binding to a subset of these proteins and they show the enzymatic activities of one histidine kinase and one diguanylate cyclase are regulated by AI-2. Finally, the authors identify ~1500 proteins from diverse organisms (including archaea) that are potential AI-2 binding receptors. Notably, these proteins, all of which have dCACHE_1 domains, are predicted to have a variety of activities that could be regulated by AI-2. In a lovely finale, the authors randomly select 19 of these dCACHE_1 proteins and demonstrate AI-2 binding to two of them using a reporter assay.

Overall, this manuscript is thorough and interesting and it provides a significant new discovery. The experiments are well controlled, and the manuscript is written in a fairly straightforward manner.

Response: The authors are very grateful for the positive feedback provided by the reviewer.

One major problem with the manuscript is that there are numerous instances of plagiarism. Hopefully, this is an oversight on the part of the authors. Whole passages, with only insignificant changes from several of the foundational AI-2 studies have been lifted and patched together in the introduction and the discussion sections. The authors have also included an exact replica of a figure panel from the original study that defined the AI-2 biosynthetic pathway without credit. This manuscript cannot be considered until these plagiarism issues, and several other points, outlined below, are addressed.

Response: We thank the reviewer for raising this important question. In the revised manuscript, we have reorganized the language of the introduction and discussion sections to address the potential plagiarism issues. The references for Fig. 1a that defined the AI-2 biosynthetic pathway have been given, including the original reference (Chen *et al.* Nature 2002, 415: 545–549) showing the AI-2 molecule that makes the boron adduct, and the original reference (Miller *et al.* Mol. Cell 2004, 15: 677-687) showing the AI-2 molecule without boron.

Major concerns

1. There are multiple incidences of plagiarism. Lines 41-56 of the introduction bear such uncanny resemblance to passages from the original AI-2 papers that the line between describing what has come before and plagiarism has been crossed. The introduction needs to be rewritten by the authors and they need to change more than a word here or there or modestly change the order of the words/sentences from the original references. While not as egregious, the same goes for the initial parts of the discussion. Phrases like “monitor ligand occupancy” etc. come straight from earlier papers. If these authors want to review other scientists’ work, they need to be careful not to lift language, but rather, use their own wording. Furthermore, regarding Fig. 1A, they need to

give a reference for the biosynthesis scheme. That exact figure comes from someone else's published work and it needs to be credited, otherwise it is plagiarism. The authors also need to include the original reference (Chen) showing the AI-2 molecule makes the boron adduct. Chemotaxis to AI-2 has also previously been demonstrated in other organisms. The authors need to credit these ideas/studies earlier in the paper than in the discussion.

Response: We thank the reviewer for raising these important questions. According to the reviewer's suggestion, we have rewritten the introduction and discussion sections by our own wording to avoid plagiarism (**Lines 41-80, 269-329**). The references for the biosynthesis scheme of Fig. 1a have been given, including the original reference (Chen *et al.* Nature 2002, 415: 545–549) showing the boron-containing AI-2 molecule, and the original reference (Miller *et al.* Mol. Cell 2004, 15: 677-687) showing the AI-2 molecule without boron. Previous studies about chemotaxis to AI-2 (Laganenka *et al.* Nat. Commun. 2016, 7: 12984; Hegde *et al.* J. Bacteriol. 2011, 193: 768-773) have also been credited earlier in the introduction section (**Lines 68-70**) and the result section before the chemotaxis assays carried out by us (**Lines 95-96**).

2. It is unclear what the co-culture biofilm results presented in Figure 3B-C contribute to the authors' message. These experiments are complex, challenging to interpret, and the results could be indirect. For example, the authors show that while WT *S. aureus* stimulates *P. aeruginosa* biofilm formation, the Δ luxS *S. aureus* does not. Does the *S. aureus* Δ luxS grow at the same rate as WT in the presence of *P. aeruginosa*? What genes does AI-2 regulate in *S. aureus* that could affect the interaction between the two species? The authors claim that the result shows that *P. aeruginosa* is detecting AI-2 made by *S. aureus*. That claim is not supported by the data. Either the *P. aeruginosa* senses AI-2 made by *S. aureus*, or, equally possible is that AI-2 mediated changes alter the interactions in biofilms. This figure and companion text should be removed. The experiment does not fit well with the other points made, and most importantly, it weakens the paper. Unlike the other experiments, which are rigorous, this one is unconvincing.

Response: We thank the reviewer for this very insightful point. As suggested, the figure presenting the co-culture biofilm results and the companion text

have been removed.

3. There are no gel images or indications as to purity levels of proteins. Please provide SDS-PAGE gel images in a supplementary figure.

Response: As suggested, SDS-PAGE gel images showing purified full-length PctA, KinD and rphK1S-Z16 have been provided as **Supplementary fig. 14**.

4. Why are the K_d 's so different in Figs 2 and 4? Also, within that context, the K_d in Fig 2 and in Fig 4D is 0.16 μM , but in Fig 4A it is 56 nM with no boron. If there was boron present, the authors show that the K_d should be 3 μM . Please reconcile.

Response: We thank the reviewer for this important point. In fact, in Fig 2b and in Fig 4d, 10 μM protein was titrated with 700 μM DPD/AI-2 (Omm Scientific) under normal conditions without boron removal. The boron content of DPD/AI-2 (Omm Scientific) is unknown and the amount of this purchased reagent is insufficient for boron removal. In order to investigate the role of boron in the binding of AI-2 to PctA-LBD, *In vitro* reactions of SAH with Pfs and LuxS proteins were performed with plasticware and borate-depleted water to obtain borate-depleted DPD/AI-2. The purified PctA-LBD was dialyzed against and diluted in the borate-depleted Tris buffer to remove boron of the protein solution as much as possible. In Fig. 4a, 1 μM protein that has been dialyzed against and diluted in the borate-depleted Tris buffer was titrated with the products of the Pfs/LuxS reaction (the AI-2 concentration is approximately 15 μM) in a borate-depleted system. Thus, results described in Fig. 4a were obtained in the absence of boron and the K_d value (45 nM) (**Fig. 4a**) is much lower than that obtained under normal conditions without boron removal (0.18 μM) (**Fig. 2b and Fig. 4d**). In contrast, when the products of the Pfs/LuxS reaction in the borate-depleted system were supplemented with 150 μM boric acid, a much higher K_d value ($>3 \mu\text{M}$) was obtained (**Fig. 4b**), indicating that the addition of boron weakens the interactions between the products of the Pfs/LuxS reaction and PctA-LBD, thus suggesting the nonborated form of AI-2 is the preferred ligand for PctA-LBD. Similar results were also obtained when the products of the Pfs/LuxS reaction without or with boron addition were titrated into TlpQ-LBD in borate-depleted buffer (**Supplementary Fig. 7a, b**).

We have modified and supplemented the relevant contents in the method (Lines 450-471) and result sections (Lines 181-191) and the figure legends of Fig. 4 and Supplementary Fig. 7 to make the results easy to understand.

Minor concerns

1. In Lines 212-220-what is the explanation for the proteins that did not bind AI-2? These are all close homologs, so why do the authors think two bind AI-2 and four others do not? This result needs some text explanation.

Response: AI-2 binding activity of the six bacterial dCACHE domains were examined using the *V. harveyi* MM32 reporter assay (Fig. 5a) and text explanation have been added in the result section (Lines 229-234). For the two dCACHE domains that show AI-2 binding in the reporter assay, we have determined the binding affinity by ITC experiments (Fig. 5b, c) (Lines 234-236).

2. In the images in 3B, the results are difficult to see (particularly the abundance of *S. aureus*). As mentioned in point 2 above, probably this set of experiments should be removed altogether. At most, these images could go in the supplement as non-merged, side by side panels that would enable the reader to interpret the data. The bar plot in C is sufficient for the main text. But again, it weakens the strong message, so better to remove all of these data.

Response: As suggested, the data presenting the co-culture biofilm results have been removed.

3. The lettering of figure panels is inconsistent. For example, in Fig 1, the lettering proceeds in the standard order (left to right, top to bottom) in Fig 2, a and d are next to each other on the top of the figure. While this is a tiny point, it will be a lot easier on the reader if the layouts are made to be consistent.

Response: We thank the reviewer for this important point. As suggested, the layouts of all figures have been reorganized in the standard order (left to right, top to bottom).

4. In the discussion, please speculate on the possibility of inter-kingdom signaling via AI-2 given the presence of dCACHE_1 domains in archaeal

proteins. Furthermore, please remove “tempted to speculate”. It is a cliché. Just speculate. It is the discussion, speculating is what a discussion is for, and moreover, it is what the authors are doing.

Response: We thank the reviewer for the insightful comment. As suggested, the possibility of inter-kingdom signaling via AI-2 has been speculated (**Lines 325-327**) and “tempted to speculate” have been removed. Furthermore, we have added more information to establish a better link between our results and previous studies in the discussion section (**Lines 293-321**).

5. There are many tense and grammatical mistakes. As an example, Line 342, “medium till OD₆₀₀ of 0.8,” should be corrected to “medium to an OD₆₀₀ of 0.8”. “Casted” is not a word.... The manuscript needs to be thoroughly edited.

Response: As suggested, “medium till OD₆₀₀ of 0.8,” have been corrected to “medium to an OD₆₀₀ of 0.8” (**Line 393**), and “Casted” have been corrected to “cast” (**Line 28**). The manuscript has also been thoroughly edited to correct tense and grammatical mistakes.

6. A 2000 page supplement is not a good idea. Perhaps the authors could provide a link to the search results and host it in a data repository.

Response: We thank the reviewer for raising this important question. In fact, the supplementary data presented in this study are the results of bioinformatics analysis but do not include any new nucleic acid and protein sequences. Thus, these data have been provided as supplementary data in .xlsx and .mas files.

REVIEWER COMMENTS

Reviewer #1 (Remarks to the Author):

This is a significantly improved manuscript and the inclusion of new data has strengthened the conclusions. The very large majority of conclusions are supported by experimental data and this manuscript represents a very significant contribution to the field of sensing and signal transduction.

However, a number of issues persist and need to be addressed.

There is an inconsistency in the present manuscript as to whether the LBDs of PctB and PctC bind AI-2 or not. The authors report that PctC-LBD binds AI-2 with a K_D of 40 micromolar (Supp. Fig. 3B). However, in Supp. Fig. 11 the authors state that "The PctC-LBD unable to bind AI-2 was used as a negative control". The same holds for Supp. Fig. 9 where PctC is classified as "unable to retain AI-2".

However, having remarked these inconsistencies, the authors have shown the dilution controls for the injection of DPD/AI-2 (Supp. Fig. 13 A) into buffer. Peaks are broad on its base at the beginning and get smaller towards the end. If the authors integrated the peaks the tendency in dilution heats would be visible. (As a general comment the integrated peak areas of the dilution controls should also be shown, revealing that controls in panels A and C are suboptimal). As for panel A this is due to the fact that a mixture of interchangeable ligands is injected and the heats measured at the beginning of the dilution may represent heats due to ligand interchange.

However, due to the nature of the ligand injected (mixture) the quality of dilution experiments cannot be improved. Considering the tendency in heats observed in the dilution experiments this referee feels that the heats observed for PctB and PctC titration with AI-2/DPD may not represent binding. This could be mentioned saying that although minor heats were observed for the PctBC titration with AI-2/DPD, in the context of the suboptimal dilution control (that cannot be changed since ligands are a mixture of interchangeable compounds) it is not certain whether these heats do indeed reflect binding.

As mentioned above, the authors should also integrate the dilution heats from Supp. Fig. 13 and present both, raw data and integrated normalized values in the lower panel.

Supp. Fig. 3 and elsewhere: Constants are given with two decimal positions; considering the magnitude of the value and errors this "precision" is not justified and it is suggested that values are rounded: for example replace 28.05 ± 2.39 micromolar with 28 ± 2 micromolar.

Supp. Fig. 3: Problems persist with ITC data. Although the authors state in the legend to Supp. Fig. 3 that dilution heats were subtracted from titration data and the corrected data were fitted, the figures show the opposite. Upper panels are titration raw data showing heat changes that are caused by the titration; these heats are the sum of binding and dilution heats. This referee has difficulties in understanding the y-axis label of the upper panel "corrected heat rates". In fact shown is what the instrument has recorded. The heat rate has been corrected for what? Ligand dilution heats are those observed following protein saturation with ligands, for example the last 5 injections in Supp. Fig. 3 C and D. These heats have to be subtracted from all titration data in a way that the lower panel only represents binding heats. If this were done as indicated by the authors, the last 5 points in the lower panels of Supp. Fig. 3 C and D should be close to zero, since protein is saturated and no binding heats are observed. However, in none of the graphs the final points in the lower panel are close to zero indicating that in these data the ligand dilution heats had not been removed.

Dilution heats in Supp Fig. 13 C are also sub-optimal which will become visible if the authors integrated the peaks. However, this may be caused by sub-optimal design. A complex between AI-2 and boric acid is injected into buffer and heat changes observed at the beginning may be due to borate dissociation from AI-2. In this referee's mind a better approach would be to titrate AI-2/borate into protein containing borate and correct with AI-2/borate into buffer/borate. It is very

likely that the magnitude of borate dissociation from AI-2 differs in an injection into buffer as compared to an injection into a protein that tightly binds AI-2/borate.

Please specify the injection volumes for the ITC experiments.

Fig. 4 and Supp. Fig. 7: the authors state that the AI-2 concentration was approximately 15 micromolar. Was this concentration also used for data analysis? To this referee it seems unlikely that injections of only 15 micromolar solution (please provide injection volumes in the experimental section) give rise to such large peaks considering that the calculated reaction enthalpy is relatively modest (approx.. 30 kJ/mol).

Minor:

Line 63: converts

Line 121: suggests that they are in about the same fold, rephrase

Supp. Fig. 2: To enhance the clarity of this figure, please label the N- and C-termini as well as the membrane-proximal and membrane distal modules

Reviewer #2 (Remarks to the Author):

The authors addressed most of the concerns raised by my review, however I remain unsatisfied with the SDS-PAGE gel images of purified proteins that the authors provided. The new images show three membrane extracted full-length proteins used to examine AI-2 mediated regulation of enzyme activity (either methylation, phosphorylation, or DGC). As expected from membrane preps, the samples are not particularly pure. I am, and readers will be, most interested in assessing the purity of samples used for ITC experiments and in vitro AI-2 binding assays as these techniques are must be performed with exquisitely pure protein. The authors repeatedly perform ITC and binding assays with various receptors (~20) and not a single result has an accompanying protein gel. Notably, there is no indication in the methods section that the authors examined the purity of their LBD purifications – did this take place? The other reviewer also had major concerns with the authors' ITC results and the lack of protein gels compounds the importance of his/her questions. This issue must be resolved before further consideration as it is crucial to underpinning the validity of the authors' claims

Response to Reviewers

We wish to begin by thanking the two reviewers for their very supportive and constructive comments. Please find our detailed responses to each of the comments below.

Reviewer #1 (Remarks to the Author):

This is a significantly improved manuscript and the inclusion of new data has strengthened the conclusions. The very large majority of conclusions are supported by experimental data and this manuscript represents a very significant contribution to the field of sensing and signal transduction.

Response: We would like to thank the reviewer for the very positive comments on our study.

However, a number of issues persist and need to be addressed.

Response: We have revised our manuscript in accordance with the reviewer's comments.

There is an inconsistency in the present manuscript as to whether the LBDs of PctB and PctC bind AI-2 or not. The authors report that PctC-LBD binds AI-2 with a KD of 40 micromolar (Supp. Fig. 3B). However, in Supp. Fig. 11 the authors state that "The PctC-LBD unable to bind AI-2 was used as a negative control". The same holds for Supp. Fig. 9 where PctC is classified as "unable to retain AI-2".

Response: We thank the reviewer for this important point. In the revised manuscript, we have more specifically described the relationship between PctC and AI-2. The sentences now are: "The dCACHE domains with a high AI-2 binding activity are depicted in blue and those with a low AI-2 binding activity are in red." and "The PctC-LBD showing low binding affinity for AI-2 was used as a negative control." in the legends of Supplementary Figs. 9 and 11, respectively.

However, having remarked these inconsistencies, the authors have shown the dilution controls for the injection of DPD/AI-2 (Supp. Fig. 13 A) into buffer. Peaks are broad on its base at the beginning and get smaller towards the end.

If the authors integrated the peaks the tendency in dilution heats would be visible. (As a general comment the integrated peak areas of the dilution controls should also be shown, revealing that controls in panels A and C are suboptimal). As for panel A this is due to the fact that a mixture of interchangeable ligands is injected and the heats measured at the beginning of the dilution may represent heats due to ligand interchange. However, due to the nature of the ligand injected (mixture) the quality of dilution experiments cannot be improved. Considering the tendency in heats observed in the dilution experiments this referee feels that the heats observed for PctB and PctC titration with AI-2/DPD may not represent binding. This could be mentioned saying that although minor heats were observed for the PctBC titration with AI-2/DPD, in the context of the suboptimal dilution control (that cannot be changed since ligands are a mixture of interchangeable compounds) it is not certain whether these heats do indeed reflect binding.

As mentioned above, the authors should also integrate the dilution heats from Supp. Fig. 13 and present both, raw data and integrated normalized values in the lower panel.

Response: We thank the reviewer for the insightful comment. As suggested, in the ITC control experiments we have presented both raw titration data and integrated normalized values in the upper and lower plots, respectively (**Supplementary Fig. 15**). Furthermore, we have added the sentence “Although minor heats were observed for the titration of PctB-LBD (**a**) or PctC-LBD (**b**) with DPD/AI-2, in the context of the suboptimal dilution control (that cannot be changed since ligands are a mixture of interchangeable compounds) it is not certain whether these heats do indeed reflect binding.” in the legend of Supplementary Fig. 3 according to the reviewer’s suggestion.

Supp. Fig. 3 and elsewhere: Constants are given with two decimal positions; considering the magnitude of the value and errors this “precision” is not justified and it is suggested that values are rounded: for example replace 28.05 ± 2.39 micromolar with 28 ± 2 micromolar.

Response: We thank the reviewer for this important point. As suggested, values of disassociation constants (K_d) for PctB-LBD, PctC-LBD, the mutants

of PctA-LBD and TlpQ-LBD and borate-depleted PctA-LBD and TlpQ-LBD (with or without added 150 μM boric acid) titrated with DPD/Al-2 or the borate-depleted LuxS products (with or without added 150 μM boric acid) have been rounded to the appropriate precision (**Figs. 4a, b, d and Supplementary Figs. 3a-b, 7 and 8b**).

Supp. Fig. 3: Problems persist with ITC data. Although the authors state in the legend to Supp. Fig. 3 that dilution heats were subtracted from titration data and the corrected data were fitted, the figures show the opposite. Upper panels are titration raw data showing heat changes that are caused by the titration; these heats are the sum of binding and dilution heats. This referee has difficulties in understanding the y-axis label of the upper panel “corrected heat rates”. In fact shown is what the instrument has recorded. The heat rate has been corrected for what?

Response: We thank the reviewer for the insightful comment. As pointed out by the reviewer, upper panels are titration raw data showing heat changes that are caused by the titration. Thus, we have corrected the y-axis labels of the upper and lower plots as $\mu\text{J s}^{-1}$ and kJ mol^{-1} of injectant, respectively in all ITC figures in the revised manuscript (**Figs. 2, 4 and 5 and Supplementary Figs. 3, 7 and 15**).

Ligand dilution heats are those observed following protein saturation with ligands, for example the last 5 injections in Supp. Fig. 3 C and D. These heats have to be subtracted from all titration data in a way that the lower panel only represents binding heats. If this were done as indicated by the authors, the last 5 points in the lower panels of Supp. Fig. 3 C and D should be close to zero, since protein is saturated and no binding heats are observed. However, in none of the graphs the final points in the lower panel are close to zero indicating that in these data the ligand dilution heats had not been removed.

Response: We thank the reviewer for raising this important question. We found that even if the heat of dilution control has been subtracted from the titration, the enthalpy at saturation could not approach zero in all of our ITC experiments. In the study by Torcato et al. (J. Biol. Chem. 2019, 294:4450–4463), when DPD/Al-2 at 800 μM was injected into 117.4 and 108

μM LsrB protein from *Clostridium saccharobutylicum* and *E. coli* K-12 MG1655, respectively, the final points in the lower panel are close to zero (Fig. 6A, C in the paper by Torcato et al.). In comparison, in our study DPD/Al-2 at $700\ \mu\text{M}$ was injected into $10\ \mu\text{M}$ protein in the ITC assays, which led us to speculate that the ratio of protein to ligands may not be optimal in our ITC assays. Indeed, when DPD/Al-2 at $700\ \mu\text{M}$ was injected into **$70\ \mu\text{M}$** protein in the ITC assays, the final points in the lower panel are close to zero after the ligand dilution heats had been subtracted from the titration (**Supplementary Fig. 3c, d**). Thus, we have repeated ALL of our ITC assays with DPD/Al-2 at $700\ \mu\text{M}$ titrated into **$70\ \mu\text{M}$** protein in the revised manuscript (**Figs. 2b-c, 4d and 5b-c and Supplementary Figs. 3 and 8b**).

Dilution heats in Supp Fig. 13 C are also sub-optimal which will become visible if the authors integrated the peaks. However, this may be caused by sub-optimal design. A complex between Al-2 and boric acid is injected into buffer and heat changes observed at the beginning may be due to borate dissociation from Al-2. In this referees mind a better approach would be to titrate Al-2/borate into protein containing borate and correct with Al-2/borate into buffer/borate. It is very likely that the magnitude of borate dissociation from Al-2 differs in an injection into buffer as compared to an injection into a protein that tightly binds Al-2/borate.

Response: We thank the reviewer for this very insightful point. As predicted by the reviewer, when the boron-free products supplemented with borate were injected into borate-depleted protein buffer supplemented with borate, dilution heats are more optimal (**Supplementary Fig. 15c**). As suggested, in the ITC assays with added borate the products of the Pfs/LuxS reaction in a borate-depleted system supplemented with $150\ \mu\text{M}$ boric acid were titrated into borate-depleted PctA-LBD or TlpQ-LBD supplemented with $150\ \mu\text{M}$ boric acid (**Fig. 4b** and **Supplementary Fig. 7b**), and the microcalorimetric data were corrected by subtracting the heats of dilution for boron-free products supplemented with borate injected into borate-depleted protein buffer supplemented with borate (**Supplementary Fig. 15c**).

Please specify the injection volumes for the ITC experiments.

Fig. 4 and Supp. Fig. 7: the authors state that the AI-2 concentration was approximately 15 micromolar. Was this concentration also used for data analysis? To this referee it seems unlikely that injections of only 15 micromolar solution (please provide injection volumes in the experimental section) give rise to such large peaks considering that the calculated reaction enthalpy is relatively modest (approx.. 30 kJ/mol).

Response: We thank the reviewer for raising this important question. In the original manuscript, DPD/AI-2 concentration in the products from reaction of SAH with Pfs and LuxS was estimated by measuring AI-2 activity in the *V. harveyi* MM32 bioluminescence assay using a standard curve obtained from known concentrations of DPD/AI-2 (Omm Scientific), a method which was also used by Yu et al. (Infect. Immun. 2013, 81:4053–4062). However, in the studies by Schauder et al. (Mol. Microbiol. 2001, 41:463–476) and Chen et al. (Nature 2002, 415:545–549), the resulting yield of DPD/AI-2 in the *in vitro* Pfs/LuxS reaction was estimated by measuring the amount of homocysteine released using mass spectral analysis (*S*-ribosylhomocysteine is converted to DPD and homocysteine by LuxS and thus the concentrations of homocysteine and DPD/AI-2 are equal in the reaction products). Considering that the mass spectrometry method is more accurate, homocysteine concentration in the reaction products were analyzed by LC-MS/MS (Agilent 1260 liquid chromatography, AB SCIEX QTRAP 6500 triple quadrupole mass spectrometer) and its concentration in the reaction products is approximately 13 μM . Thus, DPD/AI-2 concentration in the products is also approximately 13 μM (DPD/AI-2 concentration was estimated to be approximately 15 μM by *V. harveyi* MM32 bioluminescence assay). We also found that the enthalpy at saturation could not approach zero when the reaction products were titrated into **1 μM** PctA-LBD or TlpQ-LBD. We thus have repeated the ITC assays with the reaction products in the presence or absence of borate titrated into **1.3 μM** protein in the presence or absence of borate and the final points in the lower panel are close to zero after the ligand dilution heats had been subtracted from the titration (**Fig. 4a, b and Supplementary Fig. 7**). We also found that when the protein concentration increased from 1 μM to 1.3 μM in the ITC assays, the titration peaks became notably smaller (**Fig. 4a, b and Supplementary Fig. 7**). In addition, injection volumes have been provided in the experimental section

(**Lines 421, 467-468, 470**) according to the reviewer's suggestion.

Minor:

Line 63: converts

Response: As suggested, we have revised the sentence as "LuxP bound to AI-2 converts the activity..." (**Line 63**).

Line 121: suggests that they are in about the same fold, rephrase

Response: We have revised the sentence as "suggests that they are mostly in the same fold..." (**Lines 122-123**).

Supp. Fig. 2: To enhance the clarity of this figure, please label the N- and C-termini as well as the membrane-proximal and membrane distal modules

Response: We thank the reviewer for this important point. As suggested, the N- and C-termini as well as the membrane-proximal and membrane-distal modules of PctA-LBD and TlpQ-LBD are labeled in **Supplementary Fig. 2**.

Reviewer #2 (Remarks to the Author):

The authors addressed most of the concerns raised by my review, however I remain unsatisfied with the SDS-PAGE gel images of purified proteins that the authors provided. The new images show three membrane extracted full-length proteins used to examine AI-2 mediated regulation of enzyme activity (either methylation, phosphorylation, or DGC). As expected from membrane preps, the samples are not particularly pure. I am, and readers will be, most interested in assessing the purity of samples used for ITC experiments and in vitro AI-2 binding assays as these techniques are must be performed with exquisitely pure protein. The authors repeatedly perform ITC and binding assays with various receptors (~20) and not a single result has an accompanying protein gel. Notably, there is no indication in the methods section that the authors examined the purity of their LBD purifications – did this take place? The other reviewer also had major concerns with the authors' ITC results and the lack of protein gels compounds the importance of his/her questions. This issue must be resolved before further consideration as it is crucial to underpinning the validity of the authors' claims.

Response: The authors are very grateful for the positive feedback and thank the reviewer for raising the important question about protein purity analysis. We have examined the purity of the LBD purifications before these proteins were used in ITC experiments and *in vitro* AI-2 binding assays. As suggested, SDS-PAGE gel images showing purified His₆-tagged LBDs used for *in vitro* AI-2 binding assays have been provided as **Supplementary fig. 13** and SDS-PAGE gel images showing purified tag-free LBDs used in ITC experiments have been provided as **Supplementary fig. 14**.

REVIEWERS' COMMENTS

Reviewer #1 (Remarks to the Author):

The authors have responded to my comments in a satisfactory manner. This is a solid piece of research that will have a major impact in the field.

signed: Tino Krell

Reviewer #2 (Remarks to the Author):

The authors have satisfied my concerns with the new revisions.

REVIEWERS' COMMENTS

Reviewer #1 (Remarks to the Author):

The authors have responded to my comments in a satisfactory manner. This is a solid piece of research that will have a major impact in the field.

signed: Tino Krell

Response: We would like to thank the reviewer for the very positive comments on our study, and no further issues were raised by the reviewer.

Reviewer #2 (Remarks to the Author):

The authors have satisfied my concerns with the new revisions.

Response: No further issues were raised by the referee.